# Anti-CTLA4 treatment reduces lymphedema risk potentially through a systemic expansion of the FOXP3$^+$ T$_{reg}$ population

Stefan Wolf[1], Matiar Madanchi [2], Patrick Turko [2,3], Maija Hollmén [4], Sonia Tugues[5], Julia von Atzigen[1], Pietro Giovanoli[1], Reinhard Dummer [2], Nicole Lindenblatt[1], Cornelia Halin [6], Michael Detmar [6], Mitchell Levesque [2] & Epameinondas Gousopoulos [1] ✉

Secondary lymphedema is a common sequel of oncologic surgery and presents a global health burden still lacking pharmacological treatment. The infiltration of the lymphedematous extremities with CD4$^+$T cells influences lymphedema onset and emerges as a promising therapy target. Here, we show that the modulation of CD4$^+$FOXP3$^+$CD25$^+$regulatory T (T$_{reg}$) cells upon anti-CTLA4 treatment protects against lymphedema development in patients with melanoma and in a mouse lymphedema model. A retrospective evaluation of a melanoma patient registry reveals that anti-CTLA4 reduces lymphedema risk; in parallel, anti-CTLA4 reduces edema and improves lymphatic function in a mouse-tail lymphedema model. This protective effect of anti-CTLA4 correlates with a systemic expansion of Tregs, both in the animal model and in patients with melanoma. Our data thus show that anti-CTLA4 with its lymphedema-protective and anti-tumor properties is a promising candidate for more diverse application in the clinics.

Secondary lymphedema is characterized by the inefficiency of the lymphatic vasculature to drain the extravasated fluid, leading to fluid stasis and progressive fibro-adipose tissue deposition[1]. In the Western world, the iatrogenic injury of the lymphatic vasculature due to oncologic surgery and/or radiotherapy presents the initial triggering event[2]. Approximately 20–40% of all cancer survivors develop lymphedema[3], resulting in more than 90 million individuals suffering from the condition, with the number of patients increasing every year due to the prolonged life expectancy and the chronicity of the disease. The patients suffer from the swelling and subsequent restrictions in function of the affected extremity, as well as pain, recurrent local or systemic infections and decreased quality of life. In the absence of a curative treatment, the vast majority of the patients tries to mitigate the symptoms in a conservative manner using compression garments and lymphatic drainage therapy[4]. While lymphatic reconstructive microsurgery (lymphatico-venous anastomosis and vascularized lymph node transfer) is proven to efficiently reduce lymphedema in its earlier stages, a curative treatment is rarely achievable and the impact of the surgical interventions is limited by their availability in very few specialized centers only[5]. Thus, the development of a curative or preventive pharmacological treatment to enable broad patient accessibility is urgently needed.

Great progress has been made in the last decade in understanding the mechanisms underlying lymphedema onset and development. Historically, it was believed that the lymphatic vascular disruption together with the inability to establish collateral pathways to drain the fluid was defining lymphedema pathophysiology[6]. The strategies employing lymphangiogenic growth factor delivery to re-establish the

[1]Division of Plastic Surgery, University Hospital Zurich, University of Zurich, Zurich, Switzerland. [2]Division of Dermatology, University Hospital Zurich, University of Zurich, Zurich, Switzerland. [3]Bioinformatic Department, University Hospital Zurich, University of Zurich, Zurich, Switzerland. [4]MediCity Research Laboratory, University of Turku, Turku, Finland. [5]Institute of Experimental Immunology, University of Zurich, Zurich, Switzerland. [6]Institute of Pharmaceutical Sciences, ETH Zurich, Zurich, Switzerland. ✉e-mail: epameinondas.gousopoulos@usz.ch

lymphatic conduits and treat lymphedema were found though either inconclusive or inducing side-effects such as vascular leakage or being precarious for oncologic patients[7,8]. Recent research suggests that the primary lymphatic insult is merely a trigger and the non-resolving accumulation of protein-rich interstitial fluid is accompanied by a prominent lymphoid cell infiltration that leads to fibrosis development. In particular, lymphedematous tissue is characterized by a chronic T helper 2 (Th2) biased immune response, which has been shown to promote a pro-fibrotic environment with increased collagen deposition, decreased collagen breakdown and increased expression of pro-fibrotic and anti-lymphangiogenic cytokines such as TGFb, IL-4 and IL-13[9].

Chronic inflammation is a key determinant of lymphedema[10] and presents one of the most promising pharmacological targets, with various preclinical and clinical studies focusing on innate and adaptive immune pathways. In that regard, based on encouraging results from animal studies[11], Rockson et al. used ketoprofen, a well-known non-steroidal anti-inflammatory drug, in an open-label, exploratory trial in 21 patients with upper or lower extremity lymphedema[12]. Ketoprofen improved lymphedema skin pathology but failed to improve limb volume or bioimpedance[12]. Following further research, the efficacy of ketoprofen was found to be attributed to the 5-lipoxygenase metabolite leukotriene B4 ($LTB_4$) and not the cyclooxygenase (COX) pathway. Thus, the selective $LTB_4$ antagonist bestatin (Ubenimex) was then selected for a multicenter placebo-controlled trial (NCT02700529), with the results currently pending[13].

Besides the innate immune system, the $CD4^+$ lymphoid cell population presents another promising target in lymphedema, with an extensive number of pre-clinical and clinical studies focusing on the $CD4^+$ cell dynamics. In fact, the infiltration of CD4+ cells was found to correlate with the severity of the disease[14] and preclinical studies originating from independent groups could show that the depletion of $CD4^+$ cells (using CD4 knockout mice or neutralization antibodies) had a protective effect in lymphedema mouse models[15–17]. Additionally, the induction of the regulatory T cell ($T_{reg}$) population (using Il2/mAbIl2 complexes or $T_{reg}$ transplantation) was able to significantly reduce edema development and fibrotic tissue deposition and to improve lymphatic function[15]. Similarly, the macrolide calcineurin inhibitor tacrolimus, a potent $CD4^+$ cell immunosuppressor that exerts its properties by inhibiting the nuclear factor of activated T cells (NFAT), ultimately reducing IL2 expression, has been successfully applied in preclinical models[18]. Based on these results, an open-label, single-arm Phase II clinical study was conducted, indicating statistically significant, yet clinically marginal changes in limb volumes and an improved quality of life[19]. In another approach, aiming to modify the Th1/Th2 dynamics ultimately reducing the localized pro-fibrotic milieu, the blockade of interleukin 4 (Il4) and interleukin 13 (Il13) was used to modulate the Th2 differentiation of $CD4^+$ cells and could prevent and treat murine lymphedema[20]. Despite these encouraging experimental results, the open-label, phase I clinical study (NCT02494206) using QBX258, an experimental drug consisting of IL4 and IL13 neutralizing antibodies failed to improve limb volumes or bioimpedance, with the positive results limited to histological markers and quality of life indicators[21].

The aforementioned studies indicate that modulation of the T cell response can decisively influence the course of lymphedema. To enable an easier application, all studies used already FDA-approved drugs (bestatin, ketoprofen, tacrolimus) or drugs in a late clinical-stage clinical trial (QBX258). This guaranteed that the drug had a sufficient safety profile in preclinical models and humans, facilitating a quick and cost-efficient transition into the clinic. All clinical studies though emanated from successful preclinical studies, with limited or no supporting evidence of their translatability in humans. As a result, their

clinical implementation could moderately improve histological features of lymphedema but failed to reduce edema in treated patients.

Based on the well-established concept that immunomodulation influences lymphedema onset and development, we evaluate in this study the repurposing of the immune checkpoint inhibitor anti-CTLA4 in lymphedema treatment. We perform a retrospective assessment of a melanoma patient cohort, where patients receive lymphadenectomy and adjuvant immunomodulatory treatment as part of their oncologic regime. The results indicate that the patients receiving lymphadenectomy followed by anti-CTLA4 therapy have a significantly reduced risk of developing lymphedema compared to untreated patients or patients receiving a different immune checkpoint inhibitor scheme. We further confirm the clinical results in a mouse-tail lymphedema model, where anti-CTLA4 administration significantly reduces edema formation and improves lymphatic function. Importantly, we identify a consistent expansion of the $Foxp3^+$/$FOXP3^+$ $T_{reg}$ population, both in the preclinical model and in treated patients as the mediator of the therapeutic effect of anti-CTLA4. These results strongly suggest that anti-CTLA4 holds great promises as drug-repurposing candidate to prevent lymphedema onset and development.

## Results

### Anti-CTLA4 reduces lymphedema risk in melanoma patients

Melanoma patients undergoing lymphadenectomy have an increased risk of developing lymphedema and frequently receive adjuvant immunotherapy to limit tumor growth and improve long-term survival rates. Thus, they present the ideal patient pool to assess the effects (positive or negative) of immunomodulation upon the lymphatic vascular insult. For this purpose, the melanoma patient registry of the Department of Dermatology of the University Hospital Zurich was assessed and 1464 melanoma patients were retrospectively screened for the development of lymphedema, with a study structure as illustrated in Fig. 1A. Out of the 1464 patients, 461 underwent only sentinel lymph node biopsy (SLNB) without further interventions, with 13 patients (2.8%) developing lymphedema; a result in line with the published literature. Lymphadenectomy (LAD) was received by a total of 479 patients (regardless of subsequent adjuvant treatments), with 68 patients (14.2%) developing lymphedema (Fig. 1B). In order to dissect the effect of immunotherapy on lymphedema risk, we analyzed within the LAD group if immunotherapy either as monotherapy or in combination (anti-CTLA4, anti-PD1 or interferon) influences lymphedema risk. 18.8% of all patients receiving LAD without immunotherapy developed lymphedema, whereas immunotherapy cumulatively significantly reduced the lymphedema risk to 10.5% (Fig. 1C). We further studied the dynamics of each immunomodulatory treatment, and we identified remarkable differences among the different drugs. While the prevalence under anti-PD1 therapy was comparable to the prevalence observed in patients without treatment (19% anti-PD1 treatment vs. 18.8% only LAD/untreated), the anti-CTLA4 treatment showed a remarkable and significant reduction of lymphedema risk. Only 2.9% of the patients receiving anti-CTLA4 treatment post-LAD developed lymphedema, indicating a highly significant effect of anti-CTLA4 monotherapy in preventing lymphedema. Interestingly, the combination therapy of anti-CTLA4 and anti-PD1 tended to reduce lymphedema risk without reaching significance. The interferon group in all combinations showed results comparable to the untreated group and was too small to assess any differences between interferon monotherapy or the combination with anti-CTLA4 (Fig. 1C). Since lymphedema is a highly variable disease, we additionally analyzed possible confounding factors (e.g., Breslow depth, Clark level, tumor metastasis, patient gender etc) as well as the duration between LAD and the diagnosis for each treatment group without detecting any significant differences (Supplementary Tables 4–6). Furthermore, to ensure objectivity and transparency, the diagnostic characteristics, the

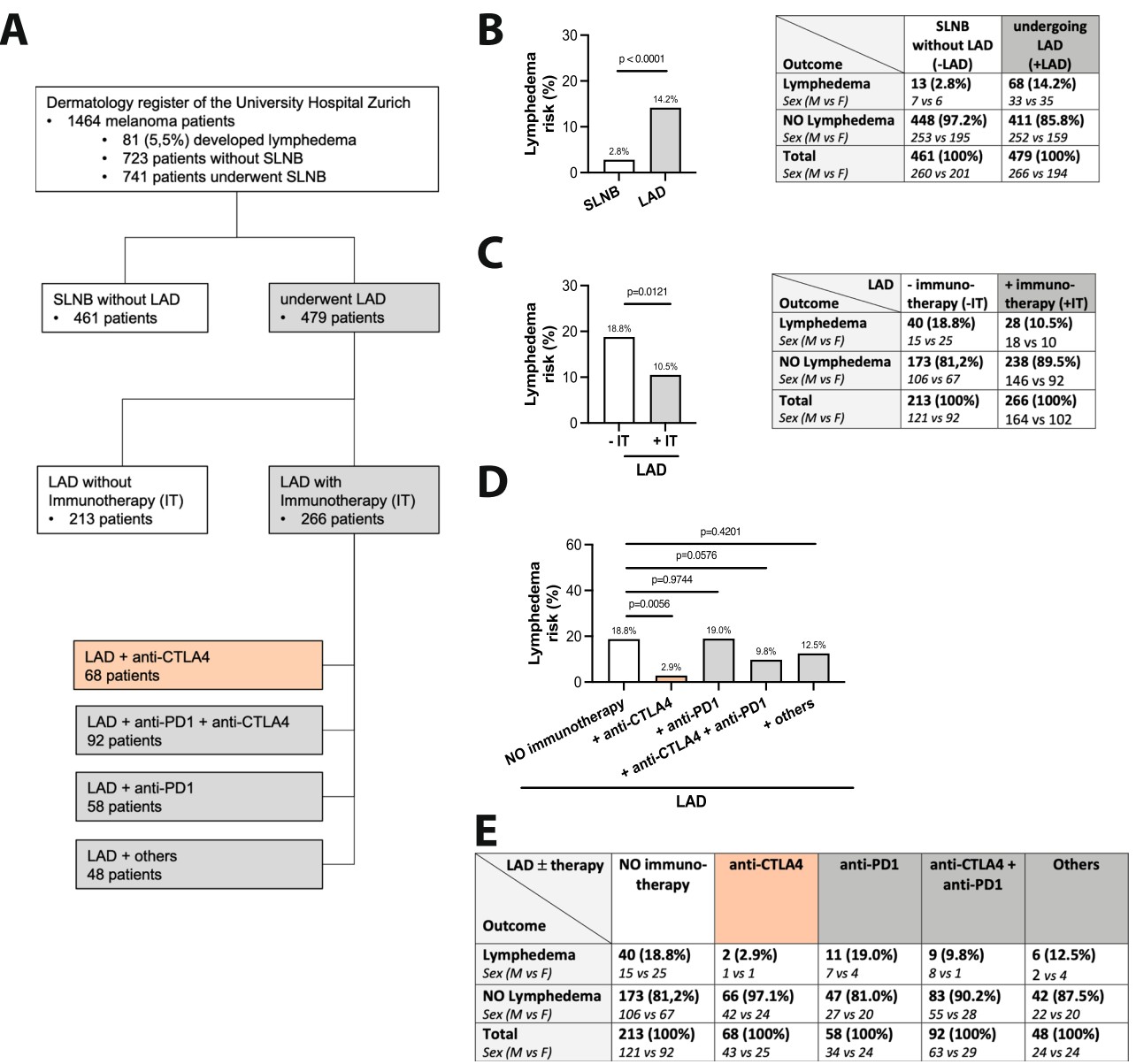

**Fig. 1 | Anti-CTLA4 therapy reduces lymphedema risk in melanoma patients.**
**A** Study flow diagram: 1464 melanoma patients were reviewed, out of which 479 underwent lymphadenectomy (LAD). 266 of the LAD treated patients received adjuvant immunotherapy, divided into 4 subgroups: anti-CTLA4 treatment (68 patients), combination treatment with anti-CTLA4 and anti-PD1 (92 patients), anti-PD1 treatment (58 patients) and interferon therapy (48 patients). **B** Melanoma patients undergoing LAD show a significantly higher lymphedema risk than patients receiving only sentinel lymph node biopsies. **C** Immunotherapy (IT) cumulatively reduces the lymphedema risk in melanoma patients undergoing LAD. **D** Anti-CTLA4 therapy reduces significantly the risk of lymphedema development upon LAD, whereas the risk in patient receiving anti-PD1 and interferon treatment is comparable to the group receiving only LAD and patient numbers are provided in (**E**). Significance was determined via Fisher's Exact Test (**B**, **C**) and for the comparison between the treatment groups a generalized linear model was used (**D**).

clinical discipline, and the diagnostic modalities used for each patient are provided in the Supplementary Table 4.

The absolute numbers of patients with or without lymphedema for each group are provided directly in the Fig. 1 B, C, E. Detailed information about the patient characteristics of the melanoma and lymphedema patient groups, the surgical procedures, and the anatomical distribution of lymphedema location, is provided in the supplementary material (Supplementary Tables 1–3).

**An increased FOXP3⁺ cell infiltration is present in lymphedema patients**
The results from the retrospective patient cohort analysis indicated that CTLA4 is a potential target molecule in lymphedema pathophysiology. Thus, we next attempted to confirm these findings in human skin

sections from secondary lymphedema and anatomically-, BMI- and age-matched control patients. Paraffin-embedded skin sections were used and stained for CD4, FOXP3 and CTLA4. $CD4^+$ (control [C]: $50.9 \pm 9.9$ cells/mm² versus lymphedema [LE] $100.3 \pm 38.0$ cells/mm²) (Fig. 2A) $FOXP3^+$ (control [C]: $4.07 \pm 2.13$ cells/mm² versus lymphedema [LE] $20.22 \pm 12.79$ cells/mm²) (Fig. 2C) and $CD25^+$ cells (control [C]: $14.36 \pm 8.12$ cells/mm²versus lymphedema [LE] $21.10 \pm 10.81$ cells/mm²) (Fig. 2B) were found to be increased in lymphedema tissue versus the matching controls. These findings were further confirmed by evaluating the mRNA expression levels of the same markers in adipose tissue, with CD25 and CTLA4 showing a twofold and 2.40-fold upregulation respectively, while the expression levels of FOXP3 remained comparable (Fig. 2D). Furthermore, we provide a lymphedema stage dependent analysis in Supplementary Fig. 1.

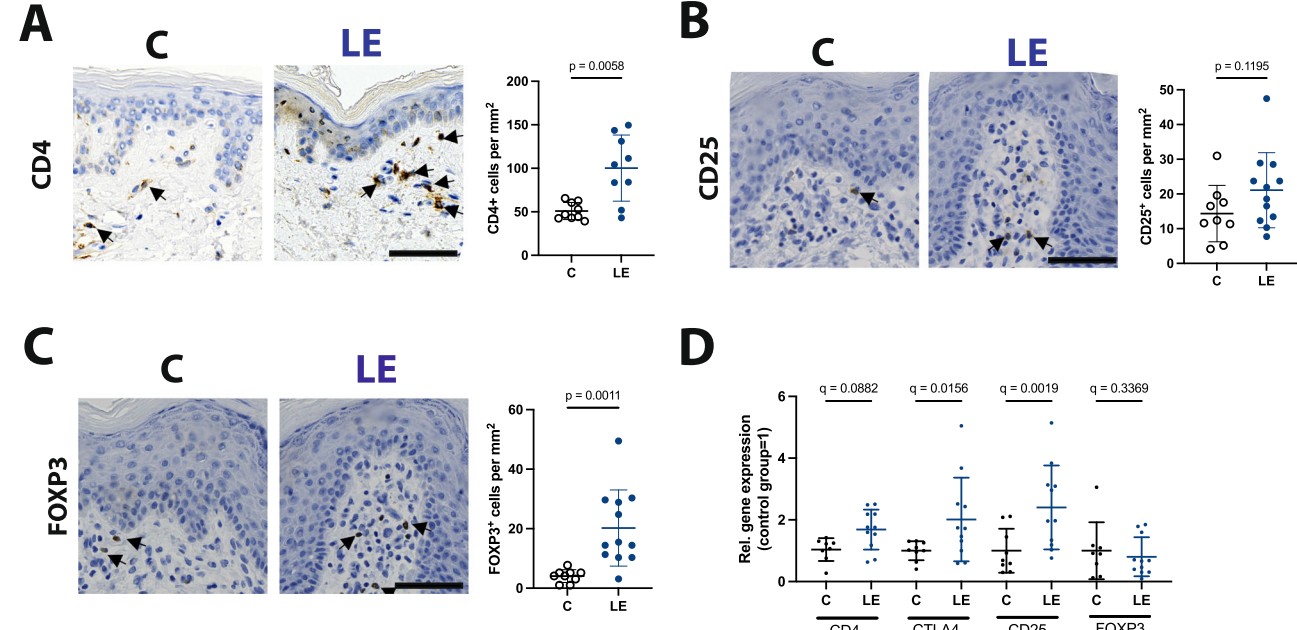

**Fig. 2 | Increased infiltration with CD4⁺, CD25⁺ and FOXP3⁺ cells, as well as increased CD4, CD25 and CTLA4 expression characterizes lymphedema tissue.** **A**–**C** Representative immunohistological images and quantification of histological analysis in paraffin skin tissue sections. An increased infiltration of CD4⁺ (**A**), CD25+ (**B**) and FOXP3⁺ (**C**) cells was detected in skin sections from lymphedema patients. Arrows indicate positive cells. **D** The evaluation of the CD4, CTLA4, CD25 and

FOXP3 mRNA expression in human subcutaneous fat tissue revealed increased CD4, CD25 and CTLA4 expression levels in patients with lymphedema. N(C) = 9 patients and N(LE) = 12 patients (**A**–**D**). Line represent mean ± SD of each group. Significance was determined via Welch's *t*-test (**A**–**C**) and ANOVA and the pairwise multiple comparison analysis was corrected for multiple comparisons the method of Benjamini, Krieger and Yekutieli (**D**). Scale bars: 200 μm.

## Anti-CTLA4 treatment exhibits efficacy in a preclinical mouse lymphedema model

Based on the results of the retrospective patient cohort evaluation, indicating that anti-CTLA4 acts protectively against lymphedema in patients, and the confirmation of increased CTLA4 expression and increased T$_{reg}$ infiltration in human lymphedema biopsies, CTLA4 was assessed as a potential target in lymphedema pathophysiology. To investigate the functional role of CTLA4 in lymphedema, we administered anti-CTLA4 mAbs in the well-established mouse-tail lymphedema model (female, BALB/c strain, 8–12-weeks-old) as of the third post-operative day (Day 3 post-operatively) in a manner comparable to the administration of anti-CTLA4 in human patients several weeks following the LAD (Fig. 3A). The treatment significantly decreased tail volumes already within 1 week, with the reduction versus the control group being consistent up to 2 weeks post operatively (endpoint) (Fig. 3B, C). It is well accepted that lymphatic hyperplasia and the subsequent functional lymphatic impairment represent distinct hallmarks in the experimental mouse lymphedema models. Thus, we next opted to evaluate the lymphatic vessel morphology and function. Tail skin sections were stained by immunofluorescence for the lymphatic marker LYVE-1. Anti-CTLA4 treatment resulted in a significant reduction of cutaneous lymphatic vessel coverage, a surrogate parameter to describe the decreased edema formation (Fig. 3E).

Furthermore, lymphatic vascular drainage function was assessed using non-invasive near-infrared (NIR) intravital microscopy as previously described[15]. The PEGylated NIR dye was injected 0.5 cm proximally to the tip of the tails and the dye migration towards the surgical excision margin was monitored and quantified. In agreement with the morphological changes observed by tissue histology, anti-CTLA4 treatment resulted in significantly improved lymphatic transport function compared to the control group, attributed to the decreased edema (Fig. 3D and Supplemental Videos 1 and 2). Furthermore, we assessed the collecting lymphatic vessels pumping capacity and aCTLA4 increased the contraction amplitude significantly

(Fig. 3F). Additionally, we evaluated the T$_{reg}$ cell infiltration in the lymphedematous tail tissue and identified an increased presence of Foxp3⁺ cells in the anti-CTLA4 treated group, in line with previous findings (Fig. 3G)[15].

Despite our focus on inhibiting lymphedema onset and in order to verify the persistence of our findings, we further examined the effect of aCTLA4 in C57BL/6 background mice, that enable longer observation periods. Female C57BL/6 mice were treated with aCTLA4 for 4 weeks post operatively, showing a significant lymphedema reduction 3 and 4 weeks post operatively (Supplementary Fig. 4 A, B) as well as a significant reduction of epidermal thickness (Supplementary Fig. 4G) and increased contraction amplitude (Supplementary Fig. 4E). Gene expression analysis of the mouse tail skin tissue confirmed once again the increased expression of T$_{reg}$-related genes such as *Foxp3* (twofold), *CD25* (1.73-fold) and *Ctla4* (2.11-fold) under aCTLA4 treatment. The comparability of the two models and thus the efficacy of the aCTLA4 treatment is presented by assessing the Area Under the Curve (AUC), as presented in Supplementary Fig. 5.

## Anti-CTLA4 treatment does not influence the Th1/Th2 immune response balance but results in a distinct PBMC composition in the BALB/c mice

As CTLA4 administration was found to have a protective function both in prospective preclinical and retrospective human studies, we next sought to assess in depth the changes induced in the local and systemic cytokine milieu upon the anti-CTLA4 treatment in the lymphedema mouse tail model.

Gene expression analysis of lymphedema tissue was performed for a number of helper T cell markers to assess the potential influence on the Th1/Th2 immune response balance. Anti-CTLA4 treatment alters the T cell composition in tumors, inducing an expansion of the ICOS+ Th1-like CD4 effector population, while outbalancing the Th2 response[22]. Thus, we analyzed the gene expression of Th1 (*Il2, Infγ, Tnfα and Cxcr3*) (Fig. 4A) and Th2 (*Il10, Il13, Ccr4 and Ccr8*) (Fig. 4B) related

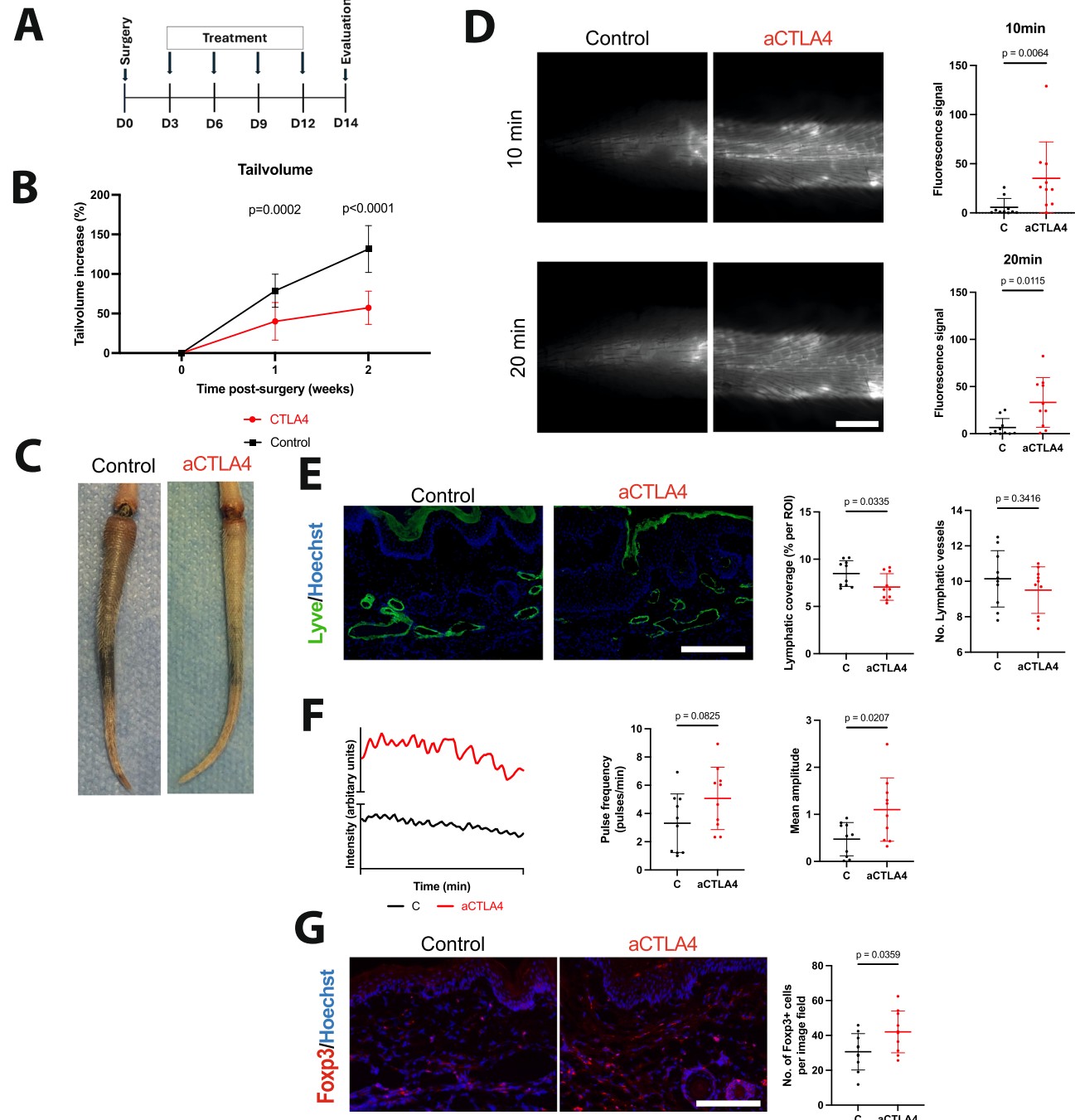

**Fig. 3 | Anti-CTLA4 treatment exhibits efficacy in the lymphedema mouse tail model. A** Surgery was performed on day 0 (D0), treatment was given on day 3, 6, 9 and 12 (D3, D6, D9, D12) and evaluation/endpoint was in day 14 (D14). **B** Anti-CTLA4 administration (aCTLA4) following the surgical induction of lymphedema leads to significantly reduced edema 1 and 2 weeks (W) after surgery. (N(C) = 10 and N(aCTLA4)=10) **C** Representative photographs of tails of control and anti-CTLA4 (aCTLA4)−treated mice two weeks postoperatively. **D** Representative images of near-infrared intravital microscopy of the tail lymphatic network -1.5 cm distally to the surgical site, visualized through the uptake and transport of a lymphatic-specific fluorescent tracer 10 and 20 min after infusion near the tip of the tail. Quantification of lymphatic vascular transport based on fluorescence intensity revealed significantly increased lymphatic vessel function in the anti-CTLA4 treatment group. Scale bar: 2000 μm **E** Representative microscopical pictures of the lymphatic vascular morphology, indicating a normalized lymphatic phenotype in the anti-CTLA4−treated group. Scale bar: 250 μm **F** Graphical representation of lymphatic vessel pulsations in the tail collecting vessels of mice treated with or without aCTLA4 2 weeks. Quantification of pulsation frequency and amplitude is shown to the right **G** Representative pictures illustrating the significantly increased Foxp3+ cell infiltration in the anti-CTLA4−treated group. Scale bar: 125 μm. All experiments show data from the same 10 control mice (**C**) and 10 aCTLA4 treated mice (aCTLA4). Line represent mean ± SD of each group. Significance was determined via a 2- way ANOVA followed by a Bonferroni's multiple comparisons test (**B**) and Welch's *t*-test (**D**, **E**, **F**).

genes. Surprisingly, we did not detect any significant alteration of either immune responses. Next, the levels of 23 cytokines were evaluated both in mouse tail lymphedema tissue and serum using Multiplex ELISA. While in the tail tissue all cytokines were detectable, only Il-1α was found significantly increased in the anti-CTLA4 treated group 2 weeks post operatively (Fig. 4E, F). In the mouse serum, the following 10 out of the 23 cytokines were detectable: Il-12 (p70), Il-17, Eotaxin, G-CSF, Infγ, KC, Ccl4, Ccl5 and TNFα. While Tnfα (control [C]:

# Tail tissue

# Serum cytokines

# PBMCs

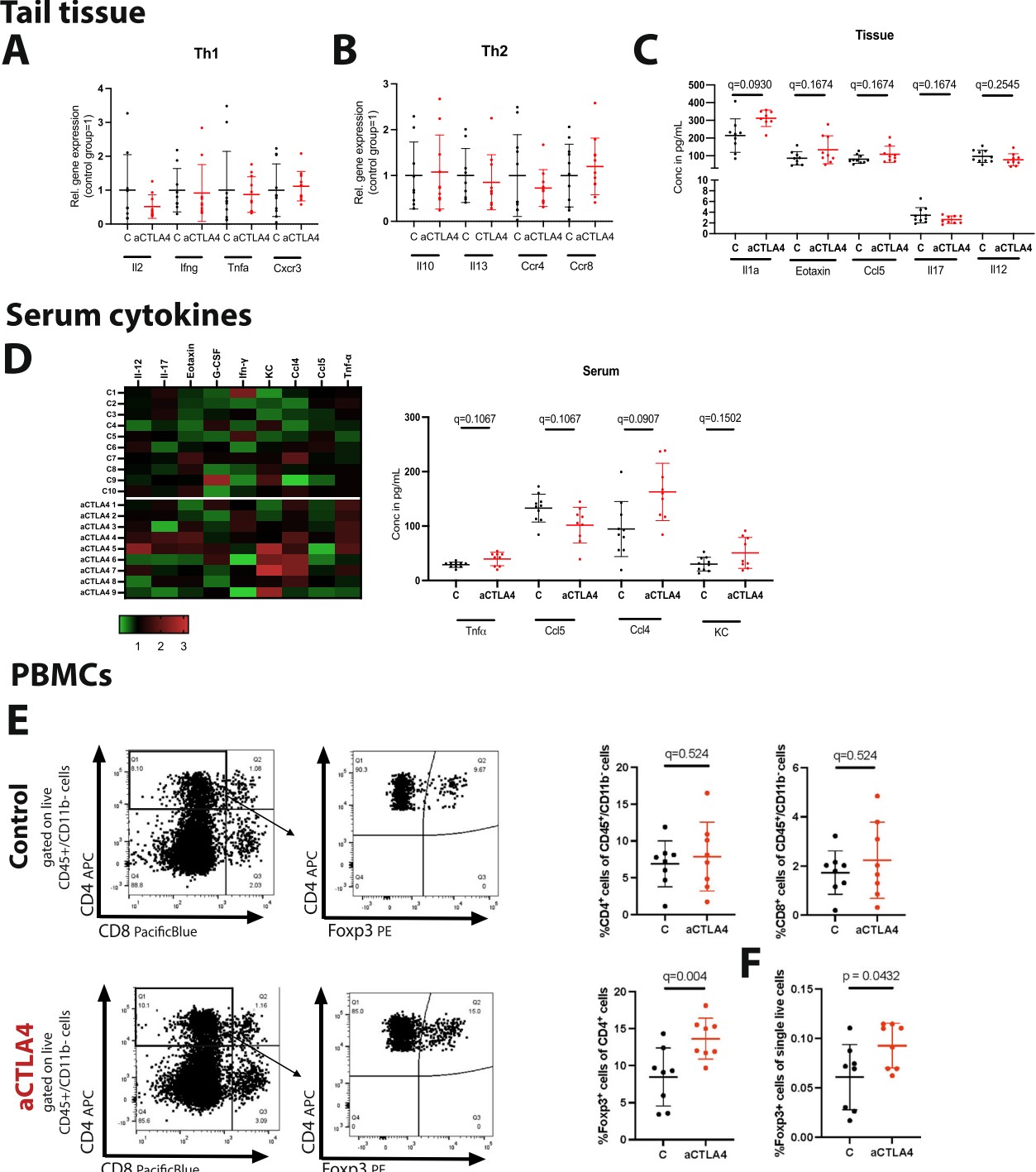

**Fig. 4 | Anti-CTLA4 treatment does not influence the Th1/Th2 balance but results in a distinct serum cytokine milieu. A, B** The evaluation of the Th1 (*Il2, Infγ, Tnfα and Cxcr3*) and Th2 (*Il10, Il13, Ccr4 and Ccr8*) marker mRNA expression in lymphedematous tail skin suggests that the Th1/Th2 immune response balance is not affected after 2 weeks of anti-CTLA4 treatment (aCTLA4) in lymphedema. **C** Quantification of altered tissue cytokines, indicating that only Il1-α increases locally in the lymphedematous tissue in response to anti-CTLA4 treatment. **D** Normalized heatmap of serum cytokines. Quantification of significantly altered serum cytokines, indicating the increased levels of Tnfα, Ccl3, Ccl4 and Il5 while the levels of Ccl5 decrease in response to the treatment. (N(C) = 10 mice and N(aCTLA4) =9 mice). **E** Representative flow cytometry analysis graphs of the phenotypic characterization performed to assess the circulating lymphoid compartment using PBMCs (gated on CD45+/CD11b−) in mice treated with anti-CTLA4 versus controls.

The analysis indicates comparable CD4+ and CD8+ cell frequencies between the two groups. Representative flow cytometry analysis graphs of the Foxp3+ T_reg compartment in murine PBMCs (gated from CD45+CD11b−CD4+) upon anti-CTLA4 treatment versus controls. The quantification of the Foxp3+ population between the two groups (gated on CD45+/CD11b−/CD4+) revealed increased circulating levels of Foxp3+ T_regs in response to anti-CTLA4 treatment. **F** Foxp3+ T_reg cell number was also increased within the total alive single cell population. N(C) = 10 mice and N(aCTLA4) = 10 mice (**A–C**), N(C) = 10 mice and N(aCTLA4) = 9 mice (D), N(C) = 8 mice and N(aCTLA4) = 8 mice (**E, F**). Line represent mean ± SD of each group. Significance was determined via a Welch's *t*-test (**F**). and ANOVA and the pairwise multiple comparison analysis was corrected for multiple comparisons with the method of Benjamini, Krieger and Yekutieli (**A–E**).

28.96 ± 4.74 pg/mL, anti-CTLA4 [aCTLA4]: 39.40 ± 12.34 pg/mL), Ccl4 (control [C]: 102.9 ± 45.69 pg/mL versus anti-CTLA4 [aCTLA4]: 162.8 ± 52.70 pg/mL) and KC (control [C]: 30.19 ± 12.56 pg/mL versus anti-CTLA4 [aCTLA4]: 50.74 ± 28.46 pg/mL) were increased in the anti-CTLA4 serum with an significant *p*-value, Ccl5 (control [C]: 133.0 ± 25.51 pg/mL versus anti-CTLA4 [aCTLA4]: 101.8 ± 32.73 pg/mL) was found to be reduced (Fig. 4C, D). Even though these results did not withstand a correction of multiple comparison, a clear trend was visible.

BALB/c mice were treated with anti-CTLA4 or control IgG in a manner faithful to the treating scheme and dosing depicted in Fig. 3A and the PBMCs were isolated on day 14. The phenotypic characterization of the murine PBMCs with flow cytometry revealed comparable frequencies of CD4+ and CD8+ cells within the CD45+/CD11b- cell population (Fig. 4E), while significantly increased numbers of Foxp3+ $T_{regs}$ cells were detected in the anti-CTLA4 treated group (Fig. 4E). The Foxp3+ $T_{reg}$ cell numbers were also increased within the total live single cell population, so that to exclude that the increase is not due to a reduction in other CD4+ cell subtypes (Fig. 4F and Supplementary Fig. 6A).

### Anti-CTLA4 treatment drives Foxp3+ $T_{reg}$ expansion and triggers distinct immune response pathways in the edematous tissue in the mouse-tail lymphedema model

In order to gain insights into the T cell dynamics following anti-CTLA4 administration, a detailed phenotypic characterization of the lymphoid compartment in the affected tail skin was performed two weeks postoperatively using flow cytometry. While the anti-CTLA4 treatment did not modify the size of the CD8+ and CD11b+ cell populations between the two groups, an increased presence of CD4+ cells, CD4+Foxp3+ CD25+ $T_{reg}$ cells, CD4+Foxp3+CD44+ and CD4+Foxp3+CTLA4+ cells was detected in the anti-CTLA4 treated group (Fig. 5A). The increased overall $T_{reg}$ infiltration was further confirmed in gene expression analyses of lymphedematous tail skin, with the detection of increased CTLA4 and FOXP3 expression levels in the anti-CTLA4 treated group (Fig. 5E). Furthermore, the FOXP3+/CD25+ $T_{reg}$ cell number was also increased within the total alive single cell population, excluding that the increase is due to a reduction in other CD4 cell types (Fig. 5D and Supplementary Fig. 6B).

To elucidate the transcriptomic features and pathways underlying the protective/therapeutic effect of anti-CTLA4 in lymphedema, we performed bulk RNA sequencing analyses of full-thickness tail skin from anti-CTLA4 treated and control mice. We identified 846 differentially expressed genes (*p* < 0.05) between the two groups. Gene Set Enrichment Analysis of the differentially expressed genes revealed downregulated keratinization and keratinocyte differentiation within the anti-CTLA4 treated tissue, a finding consistent with improved lymphedema outcome. An increased dermal keratinization leads to dermal thickening, which is a major finding in lymphedema and a parameter used to assess the effectiveness of attempted interventions[23]. The modulation of the immune response, the core function of anti-CTLA4, ranked expectedly high in the list of the pathway analysis. Importantly, the cellular responses to interferon beta (Ifnβ-) and gamma (Ifnγ-) pathways, which are both responsible for the regulation of the $T_{reg}$-related immune response were found to be increased as well (Fig. 5E, F). A respective analysis was performed to assess the molecular processes downregulated in response to the anti-CTLA4 therapy, and altered genes with a *P* value of *p* < 0.01 and log2 threshold of 0,5 were analyzed with MetaCore. In agreement with the preclinical results obtained, anti-CTLA4 therapy downregulated all key processes linked to lymphedema, namely fibrosis, epithelium degeneration and edema (Fig. 5G), further strengthening the therapeutic effect of the examined agent. These key hallmarks are all related to intestine and colon tissue, a finding that might appear confusing, but occurs as fibrosis and edema pathologies in mouse skin and soft tissue

are under-represented, thus lacking an appropriate number of reference genes. In an effort to further confirm the transferability of these findings to the lymphedema mouse tail tissue, the MetaCore enrichment ontology analysis of the toxic pathologies was applied to the RNA-sequencing results of lymphedematous tail tissue published by Gousopoulos et al.[15], indicating the same key "toxic pathologies" pathways are upregulated upon lymphedema induction (Supplementary Fig. 2).

### Anti-CTLA4 treatment drives the systemic expansion of FOXP3+ $T_{reg}$ cells in humans

The consistent expansion of the $T_{reg}$ compartment in the lymphedematous tissue upon anti-CTLA4 administration in our experimental model prompted us to evaluate whether the $T_{reg}$ expansion is not a locally-limited reaction affecting the lymphedematous tissue but a systemic response to the anti-CTLA4 treatment. For this purpose, the peripheral blood mononuclear cells (PBMCs) from anti-CTLA4 treated (monotherapy) melanoma patients were assessed by flow cytometry.

Human PBMCs were obtained from the Biobank of the Department of Dermatology originating from the same melanoma patient cohort that had been included in the retrospective study as well. Samples were collected, isolated and frozen prior to the anti-CTLA4 treatment and up to 8 weeks after treatment start. Similar to the murine experiments, the proportion of CD4+ and CD8+ cells was comparable between the two assessed timepoints in the same treated patients, but significantly increased frequencies of systemically circulating FOXP3+CD25+ $T_{reg}$ cells were detected, confirming our hypothesis (Fig. 6A–C). FOXP3+CD25+ cells were not only increased within the CD4+ population, but also with in all alive single cells. (Fig. 6D)

In order to further verify the causal relationship between anti-CTLA4 and $T_{reg}$ expansion, we next analyzed the levels of $T_{reg}$-related cytokines in the serum from the same patient cohort prior and after anti-CTLA4 treatment. In total 12 $T_{reg}$-related cytokines were evaluated in paired serum samples of 16 patients using Mesoscale U-PLEX platform. In the serum, the following 8 cytokines were detectable: MIP1b, TNFα, IL10, IL17A, IL16, IL6, INFg and CCL2. While the circulating levels of TNFα (pre: 0.74 ± 0.14 pg/mL, post: 0.96 ± 0.22 pg/mL), IL10 (pre: 0.27 ± 0.09 pg/mL, post: 0.43 ± 0.14 pg/mL), IL17A (pre: 1.18 ± 0.80 pg/mL, post: 3.32 ± 3.53 pg/mL) and IL16 (pre: 125.4 ± 32.78 pg/mL, post: 182.5 ± 79.81 pg/mL), were significantly increased 8 weeks upon the initiation of the anti-CTLA4 treatment (Fig. 6E), MIP1b, IL6, INFγ and CCL2 remained unaltered (Fig. 6F).

## Discussion

Secondary lymphedema is one of the most common complications of oncologic surgery and/or radiotherapy, yet it remains largely in the shadow as the battle against cancer and the reported survival rates dominate the spotlight[2]. However, lymphedema is a debilitating, life-long complication that while rarely fatal, can massively limit the quality of life and presents a constant reminder of its origin. Although oncologic surgeries tend to be less aggressive regarding the number of lymph nodes removed, the chronicity of the disease, coupled with the lack of curative treatments and the increased life expectancy, guarantee a steady increase in the number of lymphedema patients[24]. As such, a pharmacological approach to treat or prevent lymphedema development is urgently demanded.

The work of various research laboratories worldwide converges that the immunological determinants of lymphedema are not only a hallmark of the disease but also present potential pharmacological targets. In fact, an increasing number of publications elaborating on the role of the CD4+ cells in lymphedema suggest that immunomodulation may critically influence the onset and development of the condition. Thus far, clinical trials of potential drug candidates have been conducted based on successful preclinical studies. Their translation though

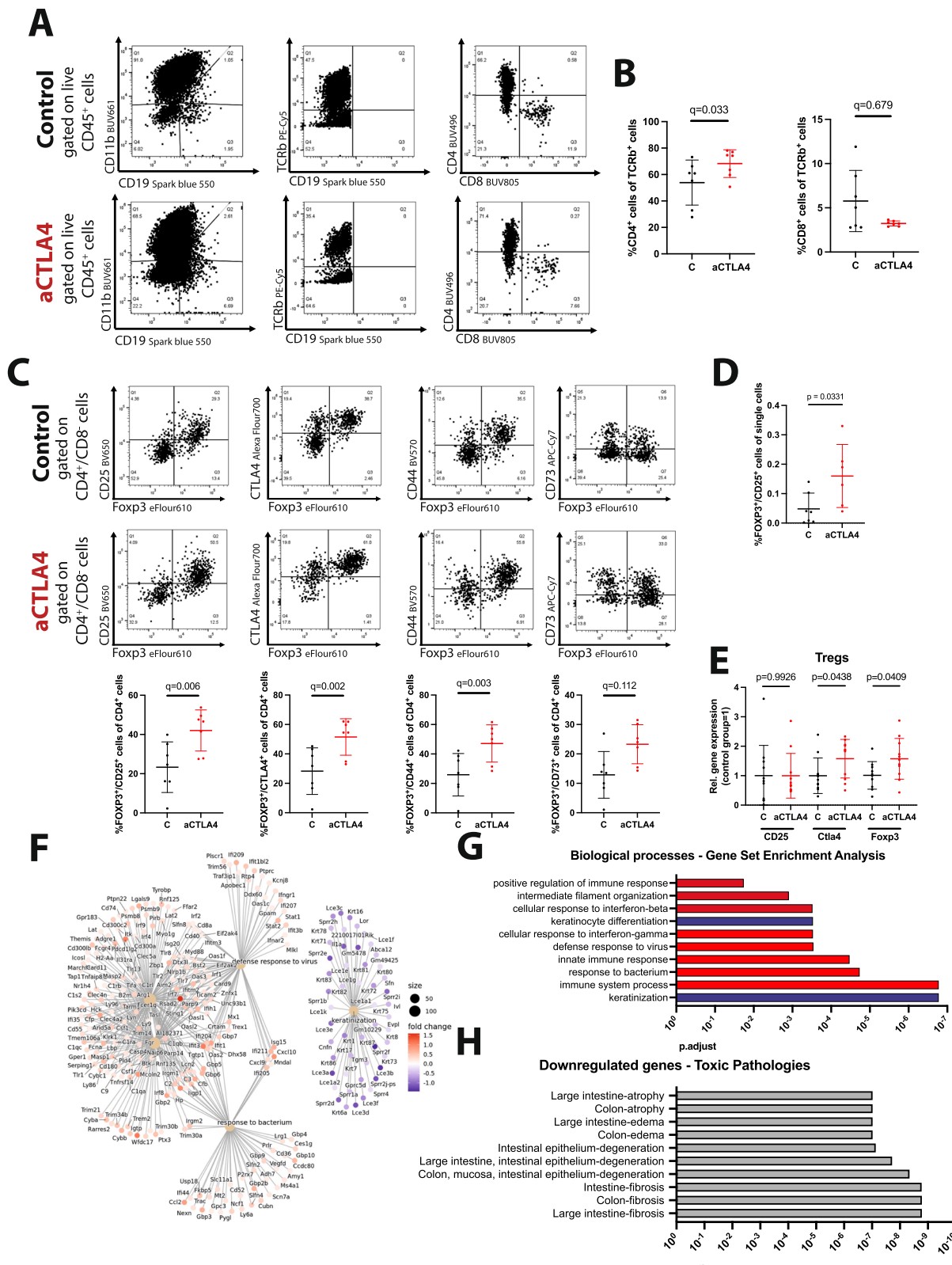

in efficient clinical interventions has been problematic and limited by the constraints of the preclinical disease models. Experimental models present irreplaceable contributors in our effort to advance research and ultimately public health, but their translatability remains often limited, as they can mimic but not substitute the human nature. This clearly indicates the paramount necessity to obtain human data in the earliest stage possible, in an effort to reduce the biological variables, assess the

relevance to the human condition and the amount of evidence obtained in humans supporting the pre-clinical outcomes.

As immunomodulation presents a promising therapeutic strategy against lymphedema, supported by a plethora of preclinical and clinical observations, we identified the melanoma patient group that receives lymphadenectomy and adjuvant immunotherapy (checkpoint inhibitors) as the most appropriate patient cohort to determine the

**Fig. 5 | Anti-CTLA4 treatment drives Foxp3⁺ T_reg expansion and triggers distinct immune response pathways in edematous mouse tail tissue.**
**A**, **B** Representative graphs of the T cell phenotypic characterization using flow cytometry. Upon initial gating for the CD45⁺CD19⁻CD11b⁻ populations, the frequency of the CD4⁺ and CD8⁺ cells were assessed, indicating an isolated CD4⁺ cell increase in response to anti-CTLA4 treatment. **C** Representative graphs of the flow cytometry analysis of the gated CD4⁺ cells. The phenotypic characterization of the T_reg suppressive function using the markers CD25, CTLA4 and CD44 cells revealed a significantly increased proportion of Foxp3⁺CD25⁺, Foxp3⁺CD44⁺ and Foxp3⁺CTLA4⁺ immunosuppresive T_regs within the otherwise expanded CD4⁺ compartment (N(C) = 7 mice and N(aCTLA4) = 7 mice). **D** Foxp3⁺CD25⁺ T_reg cell number was also increased within the total live single cell population. **E** An increased expression of *Ctla4* and *Foxp3* is present upon anti-CTLA4 treatment

while the *CD25* expression levels remained unaltered. **F** Network plots of Gene Set Enrichment Analysis showed differentially expressed genes associated with an increased immune response and reduced keratinization. **G** Analysis of activated biological processes in the anti-CTLA4 treated-group, based on Gene Set Enrichment Analysis of the differentially expressed genes. Red represents upregulated biological processes and blue are downregulated biological processes. **H** List of the 10 most significantly downregulated "toxic pathologies" as described by MetaCore enrichment analysis in the anti-CTLA4 treaded group (bulk RNA Seq: N(C) = 10 mice and N(aCTLA4) = 10 mice). Line represent mean ± SD of each group. Significance was determined via a Welch's *t*-test (**D**, **E**) and ANOVA and the pairwise multiple comparison analysis was corrected for multiple comparisons with the method of Benjamini, Krieger and Yekutieli (**B**, **C**).

clinical relevance and strength of our hypothesis, using a retrospective study. At first, we examined and validated that our estimated lymphedema prevalence corresponds to the reported findings and then proceeded with assessing the effect of immunotherapy. Melanoma patients undergoing LAD have a well-documented lymphedema risk which lies at about 20%[25,26]. Importantly, Friedman et al. investigated a cohort of 269 melanoma patients undergoing LAD and found a prevalence of 20.8%[26], which is in accordance with the 18.8% detected in our population. To our surprise though, immunotherapy (all treatment schemes cumulatively) significantly reduced lymphedema risk. Interestingly, Friedman et al. reported a related observation and highlighted a patient subgroup treated with adjuvant immunotherapy, which exhibited a lower lymphedema risk in comparison to the other patient subgroups. A more detailed analysis of the type of immunotherapies involved and the lymphedema risk for each of them is unfortunately not available[26]. Examining the different immunotherapy treatments present in our patient cohort, we were able to attribute the lymphedema protective effect to the anti-CTLA4 monotherapy, which reduced lymphedema risk to the level of the lymph node biopsy risk group. Interestingly, the combination of anti-CTLA4 and anti-PD1 did not reach the effectiveness of anti-CTLA4 monotherapy, despite the fact the same amount and frequency of anti-CTLA4 antibodies was administered in both the combination and monotherapy regimes. This effect could be mediated via the anti-PD1 treatment, that not only leads to an activation of cytotoxic T cells but to a downregulation of FOXP3 expression in the T_reg population as well, as demonstrated in the study of Wang et al. using PBMCs from melanoma patients[27].

CTLA4 is highly expressed on the surface of the T_reg cells[28], which is a substantial proportion of the lymphoid cell infiltrate in the lymphedema mouse-tail model[17] and in human samples[29]. Previous work of the senior author demonstrated the increased expression of Foxp3 and Ctla4 during the course of lymphedema development, using the mouse-tail lymphedema model and RNA sequencing of lymphedematous tissue two weeks and six weeks postoperatively[15]. With an increasing interest to identify a potential mechanistic involvement of T_regs in lymphedema, Zampell et al. attempted T_reg depletion using anti-CD25 antibodies, but the receptor CD25 is expressed on both T_regs and all activated lymphocytes, thus having collateral effects, which may explain the lack of significant effects in lymphedema[17]. Gousopoulos et al. examined the targeted role of the T_reg population in lymphedema in a number of loss-of-function and gain-of-function studies. Depletion of the T_regs was found to exacerbate edema formation and lead to rapidly increased immune cell infiltration in the affected tissue. On the contrary, the constitutive increase of the T_regs using Il2/anti-Il2 antibody complexes protected from lymphedema development, significantly reducing edema, and was associated with a reduced inflammatory burden. Similarly, the therapeutic adoptive transplantation of T_regs could reverse the pathological manifestations of lymphedema, decreasing fibrosis, inflammation and exhibiting improved lymphatic vascular function[15]. Finally, using an in vitro setup of lymphatic filariasis, Wammes et al. noted that in vitro depletion of

T_regs leads to increased Th2 pro-fibrotic cytokine responses[30]. These data vividly demonstrate the critical function of the T_regs both in the onset and course of lymphedema, a function linked to their immunosuppressive role[31]. Importantly, consistent with previous reports and the experimental models, we could demonstrate the increased FOXP3⁺ and CTLA4⁺ cell infiltration in human lymphedema biopsies and the increased CTLA4 expression in the subcutaneous tissue of secondary lymphedema patients.

Inspired by the encouraging results of the retrospective patient cohort analysis as well as the increased presence of CTLA4⁺ cells in lymphedema patients, we investigated the anti-CTLA4 treatment in the well-established murine-tail lymphedema model. Anti-CTLA4 treatment was effective in the experimental model, leading to significantly decreased lymphedema and improved lymphatic function. Furthermore, and to our initial surprise, the treatment led to an expansion of the Foxp3⁺ cell population both in the lymphedematous tissue and systemically. The transcription factor Foxp3 is known to regulate the expression of *Ctla4*[32] and previously it was hypothesized that Ctla4 blockade depletes T_regs via an antibody-dependent cell-mediated cytotoxicity[33]. More recent studies though revealed that T_reg depletion is highly dependent on tumor-infiltrating macrophages expressing the Fc receptor[34]. However, the exact effect on Foxp3⁺ cells within the tumor microenvironment remains elusive. Several studies evaluated the effect of CTLA4 blockade with controversial results: some reported a decrease in the frequency of circulating or intratumoral T_regs[34–38], while other studies did not report a T_reg depletion but even suggested a T_reg expansion following anti-CTLA-4 treatment[39–44]. In particular, Marangoni et al. reported that anti-CTLA4 therapy expands the T_reg population in a CD28-dependent manner[45]. It is important to highlight that the effect of anti-CTLA4 treatment has been investigated in depth in the context of tumor immunology, remaining unclear how precisely these findings can be transferable to the lymphedema model.

While detecting a systemic T_reg expansion upon anti-CTLA4 treatment can explain the effectiveness of the intervention, we further aimed to assess the cytokine milieu developed in response to the treatment, both locally and systemically. Indeed, anti-CTLA4 induced a distinct cytokine environment attributed to the activated immune response, as demonstrated in the RNA sequencing pathway analysis, and was found to be strongly associated with an increased Foxp3⁺ population. Il1 is known to expand the Foxp3⁺ T_reg population in vitro[46], a finding aligned to the increased Il1a concentration detected in the anti-CTLA4 treated tail skin. Similarly, the upregulated interferon beta and gamma pathway observed in the RNA sequencing pathway analysis could be attributed to the anti-CLTA4-stimulated T cells[47,48] Importantly, interferon gamma is an important Th1 proinflammatory cytokine, which can induce the Foxp3⁺ cell proliferation[49], thus actively supporting host immune responses. Interferon beta pathway was recently identified to restore both 15-LO expression and T_reg cell number in a mouse model of lymphedema[50].

In this study, the concept of immunomodulation using anti-CTLA4 antibodies was examined retrospectively in a dedicated patient

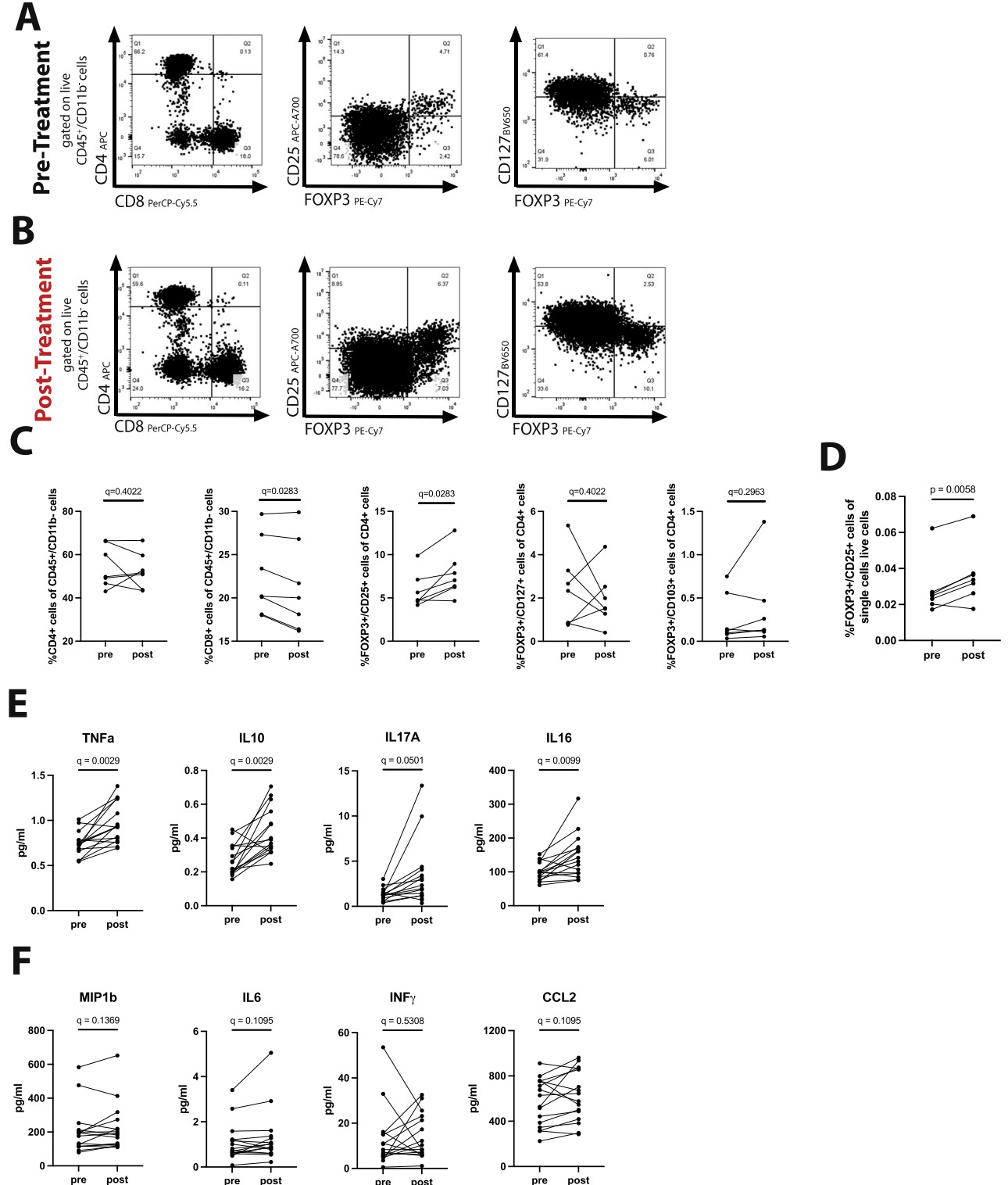

**Fig. 6 | Anti-CTLA4 treatment leads to the systemic expansion of FOXP3+ T_reg cells in humans. A** Representative flow cytometry analysis graphs of the phenotypic characterization performed to assess the circulating lymphoid compartment from human PBMCs (gated on CD45+/CD11b− cells) in melanoma patients at Day 0 (Pre-Treatment) and **B** up to 8 weeks after anti-CTLA4 treatment start (Post-Treatment). **C** Quantification of the circulating CD4+, CD8+, FOXP3+/CD25+ and CTLA4+ cells in melanoma patients at Day 0 versus Day 40 indicate an isolated systemic expansion of the FOXP3+ T_regs, induced by the anti-CTLA4 treatment.

**D** Increased percentage of FOXP3+CD25+ within all alive single cells upon aCTLA4 treatment. **E, F** Quantification of T_reg-related serum cytokines indicated increased levels of TNFα, IL10, IL17A and IL16, while the levels of MIP1b, IL6, INFγ and CCL2 remained unaltered in response to the treatment. PBMCs from 7 patients pre and post treatment (**A–D**) and serum from 16 patients pre and post treatment (**E, F**) were analysed. Significance was determined via Welch's *t*-test (**D**) and RM one way ANOVA and the pairwise multiple comparison analysis was corrected for multiple comparisons with the method of Benjamini, Krieger and Yekutieli (**C, E, F**).

cohort to substantiate the proof of concept regarding the intervention's efficiency. This was considered a critical step for the authors, as the proof of an effect in a patient population outweighs preclinical study findings and supports a more direct translational pathway. With the clinical data suggesting that anti-CTLA4 can efficiently prevent lymphedema onset in patients with lymphadenectomy, we prompted the implementation of the treatment in the mouse-tail lymphedema model, to prospectively verify the outcome and investigate the pathomechanisms underlying the observed effect. The successful results cemented the efficacy of the intervention and suggested that lymphedema improvement is mediated through an anti-CTLA4-dependent $T_{reg}$ expansion both locally and systemically. The systemic $T_{reg}$ expansion in response to the anti-CTLA4 treatment could also be verified in patients from our retrospective patient cohort, confirming the relevance of the proposed involved mechanism and the consistency between the experimental and clinical findings.

While the anti-CTLA4-mediated expansion of the $T_{regs}$ offers a solid explanation of the protective function of the intervention against lymphedema, the underlying molecular players remain unclear, as our understanding of the immune checkpoint inhibitors is still incomplete. Further studies should investigate the molecular pathways involved in the anti-CTLA4-mediated $T_{reg}$ expansion and the functional capacities of the induced $T_{reg}$ population.

The major advantage of the proposed therapeutic approach is that anti-CTLA4 is a well-established, FDA-approved oncologic medication used across the vast majority of solid tumor indications that may necessitate a lymphadenectomy (namely breast, renal and cervical cancer or melanoma), which increases lymphedema risk. Contrary to traditional anti-inflammatory therapies such as corticosteroids and non-specific immunosuppressants that bear the risk of cancer recurrence, the anti-CTLA4 treatment seems to have a dual beneficial effect for cancer patients. On the one hand, it presents a highly effective immunotherapy strategy for a large spectrum of solid cancers, and on the other hand it simultaneously offers a highly desired protective effect against lymphedema.

The number of lymphedema patients is increasing steadily due to the lack of curative treatment options and the chronic character of the disease. Thus, the pressing need to develop pharmacological approaches and the achievable impact of a potentially successful implementation is evident. While drug development is a cost-intensive and time-consuming process, drug repurposing—as shown with the anti-CTLA4 treatment approach in this study - is an emerging strategy for identifying new uses for approved drugs outside of the scope of original medical indication. An FDA-approved medication has already been found to be sufficiently safe in preclinical models and humans, meaning that the costs and time of bringing a repurposed drug to the market is a fraction of that of a new drug[51].

The current study displays certain limitations: even though the mouse-tail lymphedema model recapitulates the major hallmarks of acquired lymphedema, it cannot replicate all features, as it is restricted by the animal size and the associated anatomical limitations. The limitations of the experimental model are counterbalanced not only through the integration of advanced phenotypic, functional, and RNA sequencing methodology to study in-depth the experimental model but importantly through the adjunct retrospective melanoma patient study, which is clinically highly relevant and aligned with the experimental studies. Additionally, the usage of patient material (tissue and liquid biopsies) to verify the proposed targets and mechanisms of actions strengthens both the study design and the translatability of the obtained results. Admittedly, the treatment period of 2 weeks in the BALB/c mouse model might appear relatively short but was necessary from an animal welfare perspective, due to the prominent edema formation in the mice of the control group. This was though counterbalanced by an additional study using the C57BL/6 mice, enabling an observation period of

4 weeks upon treatment initiation, a timeframe in line with most published literature. A further extension of the course of the disease in the preclinical model would not assist the answer of our clinical question either, as the scope of the experiment was not the treatment of advanced lymphedema but to replicate the conditions to assess the protective effect of anti-CTLA4 against lymphedema onset, which was faithfully conducted with the applied setup.

## Methods

### Retrospective analysis of melanoma patients

**Study population.** The medical records from 1464 melanoma patients treated between 2011 and 2019 in the Department of Dermatology at the University Hospital of Zurich were retrospectively reviewed for the development of lymphedema. Therefore, patient details such as demographics, co-morbidities, melanoma history, and characteristics including the performance of lymph node biopsy or lymphadenectomy and adjuvant treatment with an immune checkpoint inhibitor were analyzed. For the treatment with the immune checkpoint inhibitors, the following treatment regimens were applied: anti-CTLA4 (ipilimumab) or anti-PD1 (pembrolizumab or nivolumab) were administered intravenously every 3 weeks for a total of 4 doses. If a combination was used, the anti-PD1 treatment was applied first, followed by the anti-CTLA4 treatment for 4 doses. Patients that discontinued the treatment were not included in the analysis. The lymphedema diagnosis was attributed by a dermatologist, plastic surgeon or angiologist based on clinical assessment and imaging data and the diagnosis was followed by the prescription of compressive garments and/or drainage massage. The presence of a seroma, fistula or any other differential diagnosis to lymphedema was explicitly excluded. Patient characteristics are provided in Supplementary Table 3.

**Peripheral blood mononuclear cells (PBMCs) isolation from melanoma patients.** Blood from melanoma patients, subjected to anti-CTLA4 (Ipilimumab, YERVOY, Hersteller Bristol-Myers Squibb) therapy, was collected prior to the initiation of the treatment and up to 8 weeks after the first anti-CTLA4 infusion using Vacutainer EDTA tubes. The injection scheme of each individual patient is provided in Supplementary Table 12. Blood was diluted with an equal volume of PBS and subsequently was layered carefully over Ficoll in a centrifuge tube without intermixing, followed by centrifugation at $400 \times g$ for 30 min at 20 °C. PBMCs and platelets were collected from the interface between plasma and Ficoll layers. To remove platelets, PBMCs were centrifuged in PBS at $300 \times g$ for 10 min at 20 °C. Patient details are provided in Supplementary Table 12.

**Peripheral blood mononuclear cells (PBMCs) characterization by flow cytometry.** Frozen PBMCs were quickly thawed, washed with PBS and blocked using Human TruStain FcX (422301, Biolegend). Cells were incubated with antibodies for 25 min at 4 °C listed in Supplementary Table 11 for 25 min at 4 °C. Following surface staining, cells were fixed with FOXP3/transcription factor fixation/permeabilization solution (00-5521-00, eBioscience) for 40 min at 4 °C and subsequently intracellular staining was performed. For the acquisition a Cytoflex S (Beckman Coulter) apparatus was used.

**Quantification of human serum cytokines.** Serum from the same study cohort were analyzed pre and post anti-CTLA4 (Ipilimumab) therapy using the U-PLEX Assay Platform (K151AEM-1, Mesoscale Diagnostics). Biotinylated capture antibodies are coupled to U-PLEX Linkers, which self-assemble onto unique spots on the U-PLEX plate. Analytes in the sample bind to the capture reagents; detection antibodies conjugated with electrochemiluminescent labels bind to the analytes in order to form the sandwich immunoassay. Following sandwich immunoassay completion, the U-PLEX plate is loaded into an MSD instrument. Voltage applied to the plate electrodes causes the

captured labels to emit light. The instrument measures the intensity of emitted light (which is proportional to the amount of analyte present in the sample) and provides a quantitative measure of each analyte in the sample. Each run was performed in duplicate. A customized U-PLEX assay for the analysis of the following cytokines/chemokines was used: MIP1b, TNFα, IL10, IL17A, IL16, IL6, INFg and CCL2.

### Analysis of lymphedema tissue samples

Patient characteristics of the study cohort are shown in Supplementary Table 4. The tissue collection and preparation of lymphedema samples for immunohistochemistry an

**Human tissue samples.** Skin tissue specimens were harvested during the operating procedure from lymphedema patients undergoing lymphatic reconstructive surgery or liposuction. Samples were directly fixed in paraformaldehyde/phosphate-buffered saline at 4 °C and were subsequently paraffin embedded cut into 5µm-thick sections. Sample preparation and staining was performed by the Center for Surgical Research of the University Hospital Zurich.

**Human skin immunohistochemistry.** For the immunohistochemical staining, paraffin-embedded sections were deparaffinized and rehydrated. For antigen retrieval Target Retrieval Solution high was used and endogenous peroxidase activity was inactivated with 3% hydrogen peroxide (1.07209.0250, Merck). Following blocking with goat serum, the sections were incubated with monoclonal mouse anti-human CD4 (IR649, Dako) monoclonal rabbit anti-human FOXP3 (ab215206; 1:100, abcam) or monoclonal rabbit anti-human CD25 (abcam, ab231441; 1:100) antibodies. Primary antibodies were visualized using peroxidase-conjugated anti-rabbit EnVision+ reagent (K4003, Agilent) and after PBS washing steps with DAB substrate (K3468, Chromogen), following the manufacturer's instructions.

Images were acquired with a Zeiss Axio Scan Z1 equipped with a Hitachi HV-F202FCL camera and the whole tissue section was scanned using a Plan Apochromat 20x/0.8 numerical aperture objective. For the correct and unbiased identification of FOXP3$^+$, CD4$^+$ and CD25$^+$ cells were quantified in an automated fashion using QuPath (Version 0.5.1). Patient characteristics are shown in Supplementary Table 7.

**RNA extraction and qRT-PCR analysis from human fat tissue.** Fat tissue collected during the operating procedure from lymphedema and control patients was immediately flash frozen in liquid nitrogen. From 100 mg fat tissue, the RNA was isolated using the RNeasy Lipid Tissue Mini Kit (74804, Qiagen) and subsequently cDNA was transcribed from 500 ng RNA, using the High-Capacity cDNA Reverse Transcription Kit (4368814, ThermoFisher Scientific). Quantitative PCRs were performed using Fast SYBR Green Master Mix (4385612, Applied Biosystems) and QuantStudio 5 Real-Time PCR System. B2M was used as housekeeping gene, and fold changes of gene expression were calculated using the ΔΔCT method. Primer were individually synthesized by Microsynth and sequences are listed in Supplementary Table 8.

**In vivo experimentation using the mouse tail lymphedema model**
**Surgical induction of lymphedema.** To examine the underlying pathologic processes and potential treatment approaches in lymphedema, a well-established surgically-induced mouse tail lymphedema model was employed. 8- to 12-week-old BALB/c (BALB/cByJRj, Strain Number: 001026, Janvier Labs) or C57BL/6 (C57BL/6JRj, Strain Number: 000664, Janvier Labs) mice (Janvier Labs) under specific pathogen–free housing conditions were used. The animal room had a controlled 12/12-h light/dark cycle (lights on at 6:00 AM), temperature (22 ± 2 °C), and humidity (40%–70%). Only female mice were used because male mice have higher tendency of attacking each other's tails after the surgery. Treated and Control mice were kept in pairs and

treatment for each mouse was selected randomly. A sample size of seven to ten animals per group was chosen based on our previous experience with this mouse model.

The mice were operated under Isoflurane anesthesia and 0.1 mg/kg Buprenorphine (Temgesic, Indivior Europe Limited) was injected 20 min prior to the initiation of the operation. To induce lymphedema, a 3–4 mm circumferential portion of skin was removed 2 cm distal to the tail base. The underlying collecting lymphatic vessels were visualized by injecting 5 µl of a 1,5% Evans Blue solution (E2129, Merck) and subsequently ablated under a dissecting microscope. Lymphedema developed within one week and the study was concluded at 2 weeks, with biometric data and tissue collection.

Mice will be euthanized upon anesthesia (Ketamine (100 mg/kg, Ketalar, Pfizer)/Xylazine (10 mg/kg, Rompun 2%, Bayer)) and subsequent neck dislocation.

Treatment administration: 100 µg InVivoPlus anti-mouse CTLA4 (Cat. Nr. BP0131, clone 9H10, BioXCell) or the corresponding InVivoPlus polyclonal Syrian hamster IgG control antibody (Cat. Nr. BP0087, BioXCell) diluted in InVivoPure pH 7.0 Dilution Buffer (Cat. Nr. IP0070, BioXCell) were administered on day 3, 6, 9 and 12 after surgery through intraperitoneal injection.

Exclusion criteria were preestablished and mice were excluded from analysis if any of the following incidents occurred upon surgery: (1) self-inflicted mutilation or severe abrasion on the skin, (2) severe infection and (3) loss of blood supply in the tail.

**Mouse serum collection.** Blood was collected with heart puncture upon administration of a lethal dose of anesthesia (Ketamine (100 mg/kg, Ketalar, Pfizer)/Xylazine (10 mg/kg, Rompun 2%, Bayer)) using serum micro collection tube with gel barrier (Cat. Nr. 41.1500.005, Sarstedt). Blood was incubated for 15 min at RT and centrifuged for 10 min at 1000 × g. Serum was aliquoted and stored at −80 °C until usage.

**Tail volume quantitation.** Tail volumes were measured weekly using a digital caliper from a researcher blinded to the treatment group of the mice. The diameter was measured in 1 cm intervals starting immediately distal to the surgical excision site and calculated using the truncated cone approximation[52].

**In vivo imaging of lymphatic vessel function.** Lymphatic vessel transport was assessed by a Axio Zoom V16 stereomicroscope (Zeiss) equipped with a Prime BSI Express sCMOS Camera (Teledyne Photometrics) and a pE-4000 LED light source (CoolLED). NIR dye was administered with a PHD ULTRA CP syringe pump (Harvard Apparatus) and an intradermal catheter and a 30-gauge needle was used. The lymphatic transport function was evaluated via a controlled infusion of 5 µl of the NIR dye P20D800 (20 kDa PEGylated IRDye800)[52] with a rate of 0.5 µl/min for 10 min at a site 1 cm proximal to the tail tip. Simultaneously, a video of the region ~1 cm distally to the surgical excision margin was recorded for 20 min. To evaluate the functionality of the lymphatic vessels, fluorescence intensity was measured at $t = 10$ min and 20 min in a selected region of interest 1.5 cm distal to the surgical excision margin using ZEN 3.3 software (Zeiss).

**Mouse skin histology.** During tissue collection, skin biopsies from the lymphedematous tails were embedded in molds with OCT Compound and were frozen over liquid nitrogen. Immunofluorescence staining was performed on OCT-frozen sections. 7-µm frozen sections were fixed in −20 °C acetone, 4 °C methanol and were subsequently blocked for 2 h with 5% donkey serum with 0.2% BSA and 0.3% Triton X-100 in PBS. The sections were incubated overnight (4 °C) with rabbit anti-mouse LYVE-1 (1:100, Cat. Nr. 11-034, AngioBio) or rat anti-mouse FOXP3 antibodies (1:50, Cat. Nr. 12-5773-82, eBioscience). Upon washing in PBS, the sections were subsequently incubated for 30 min

with Alexa 488– or Alexa 594–conjugated secondary antibodies (both 1:200, Cat. Nr. A-21208 and Cat. Nr. A-11058) and Hoechst 33342 (1:1000) (all from Life Technologies).

Five images per tissue section were taken with a Leica DM 5500 Microscope. Image quantification was performed using ImageJ in a blinded fashion. For the correct identification of Foxp3$^+$ cells were counted manually using QuPath by two blinded reviewers. For the quantification of the lymphatic area, lymphatic structures including the luminal space were used.

**RNA Extraction and qRT-PCR analysis.** Full-thickness tail skin portions, located 1.5–2 cm distal to the surgical excision margin were collected at 2 weeks (day 14) post operatively and were immediately flash-frozen in RNAase free Eppendorf tubes. RNA was extracted using TRIzol (Cat. Nr. 15596026, Ambion) and samples were disrupted with a TissueLyser II (QIAGEN), followed by the isolation with NucleoSpin RNA Kit (Cat. Nr. 740955.50, Macherey-Nagel) according to the manufacturer's protocol. 1 µg RNA was used for transcription of cDNA, using the High Capacity Reverse Transcription kit (Cat. Nr. 4368814, Applied Biosystems). PCR reactions were carried out using FastStart SYBR Green master mix (4385612, Applied Biosystems) and a Quant-Studio 5 Real-Time PCR Systems (Applied Biosystems). Expression values were normalized to Rplp0 and fold changes in gene expression were calculated using the ΔΔCT method. Primer sequences are listed in Supplementary Table 9.

**Multiplex cytokine analysis.** Cytokine quantification in the tail tissue and serum was performed using Bio-Plex Pro Mouse Cytokine 23-plex Assay (IL-1α, IL-1β, IL-2, IL-3, IL-4, IL-5, IL-6, IL-9, IL-10, IL-12 (p40), IL-12 (p70), IL-13, IL-17A, Eotaxin, G-CSF, GM-CSF, IFN-γ, KC, MCP-1 (MCAF), MIP-1α, MIP-1β, RANTES, TNF-α) according to the manufacturer's instructions (Cat. Nr. M60009RDPD, Bio-Rad).

**Flow cytometry analysis of anti-CTLA4 treated mice.** For the analysis of the immune composition of the lymphedematous skin, the entire tail skin of euthanized mice was harvested directly distally to the surgical margin and a single-cell suspension was prepared. Tail tissue was minced and incubated in collagenase II (5 mg/ml, Cat. Nr. C6885, Sigma-Aldrich) and DNase I (0.05 mg/ml, Cat. Nr. 7002221, Roche) in RPMI medium (Cat. Nr. 7002272, Gibco) for 30 min under agitation at 37 °C. Cell suspensions were filtered through 70-µm and 40-µm cell strainers and resuspended in FACS buffer (2% FBS, 2 mM EDTA in PBS). 10$^6$ cells/ml were stimulated for 5 h with 100 ng/mL of PMA (Cat. Nr. P1585, Merck) and 1 µg/mL ionomycin (Cat. Nr. I0634, Merck) in the presence of 1 µL/mL GolgiPlug (Cat. Nr. 555029, BD Biosciences)/GolgiStop (Cat. Nr. 554724, BD Biosciences). Subsequently, cells were washed with PBS, and non-specific antibody binding was blocked using mouse TruStain FcX (Cat. Nr. 101319, Biolegend). Cells were incubated for 25 min at 4 °C with antibodies for the surface staining. Prior to intracellular labeling, cells were fixed and permeabilized with fixation/permeabilization solution (00-5523-00, Thermo Fisher) and stained over night at 4 °C. Data were acquired with a Cytek Aurora flow cytometer (Cytek Biosciences) and analyzed using FlowJo Software (FlowJo 10.8.1). For the identification of the positive populations fluorescence minus one controls were used.

Single-cell suspensions isolated from blood were stained for surface markers followed by intracellular staining for Foxp3 using a Foxp3/Transcription Factor Staining Set (00-5521-00, eBioscience) according to the manufacturer's guidelines. Antibodies used are listed in Supplementary Table 10. Detailed experimental methods are provided in the supplementary materials.

**RNA sequencing -Library preparation, sequencing on NovaSeq 6000 and data analysis.** Quantity and quality of the isolated RNA was determined with a Tapestion (Agilent). The Illumina Stranded Total RNA Prep with Ribo-Zero Plus (Illumina) was used in the following steps. The reverse-transcription of 600 ng RNA into double-stranded cDNA was carried out in the presence of a template switch oligo. For the PCR amplification, primers binding the random priming oligo and template switch oligo sequences, which were added to cDNA fragment during reverse transcription were used. The full-length Illumina adapters, including the index for multiplexing were added during the PCR. Ribosomal cDNA was removed by ZapR in the presence of the mammalian-specific R-Probes. Enrichment of the remaining fragment was performed with a second round of PCR amplification using Illumina adapters matching primers.

The enriched library quality and quantity were assessed with a Tapestation (Agilent). The NovaSeq workflow with the NovaSeq6000 Reagent Kit (Illumina) was used for the library preparation. For cluster generation and sequencing the NovaSeq6000 System was used with a run configuration of single end 100 bp.

From the raw reads, adapter sequences and low-quality ends were removed and fastp (Version 0.20) was used for filtering reads for low quality (phred quality <20). subsequently sequence pseudo alignment to the mouse reference genome (build GRCm38.p5) was performed and quantification of gene-level expression done using Kallisto (Version 0.46.1). For the detection of differentially expressed genes, a count-based negative binomial model implemented in the software package DESeq2 (R version: 4.1.2, DESeq2 version: 1.34.0) was applied. Genes that showed an altered expression with *q*-value < 0.05 (Benjamini and Hochberg method) were considered differentially expressed.

## Statistical analysis

Statistical analysis of the retrospective study was performed using R version 4.3.2. Comparison of lymphedema frequency between treatment groups was assessed using a generalized liner model with binomial link function. Comparisons of age, Breslow, Clark level, diagnosis, and Clark level between patients developing Lymphadema (yes/no) were performed with *t*-tests (2 groups), ANOVA (>2 groups), or Fisher's exact test (categorical variables). Comparisons of the treatment groups with regards to time period elapsed between lymphadenectomy and the diagnoses of lymphedema, method of diagnosis, and other confounders were analyzed with the same methods.

Statistical analysis of the animal study and of the lymphedema patients was performed using GraphPad Prism 9.0 and all data represent mean ± SD.

For the comparison of 2 groups with a non-Gaussian distribution a nonparametric unpaired Mann–Whitney test was performed, whereas a Welch's *t*-test was performed for Gaussian distribution. For the statistical analysis of the tail volume a 2-way ANOVA with Bonferroni post-hoc test was applied. Analysis of variance and pairwise multiple comparison analysis was corrected for multiple comparisons the method of Benjamini, Krieger and Yekutieli. Grubb's test was used for the identification and exclusion of outliers. *P* < 0.05 was considered as statistically significant.

## Study approvals

The retrospective analysis of the melanoma patient cohort as well as the PBMCs collection/ isolation of the same patients was approved by the Cantonal Ethics Committee of the Canton Zürich, Switzerland (IRB approval number: KEK-ZH-Nr. 2014-0193). The tissue collection from lymphedema and control patients was approved by the Cantonal Ethics Committee of the Canton Zurich, Switzerland (KEK-ZH-Nr: 2021-02358). All recruited volunteers provided written informed consent and did not receive a participant compensation.

All experimental procedures including animals were performed in accordance with the protocols approved by the Cantonal Veterinary Office Zurich (license number ZH 105/2021).

**Reporting summary**

Further information on research design is available in the Nature Portfolio Reporting Summary linked to this article.

## Data availability

All data associated with this study are present in the paper or the Supplementary Materials. RNA sequencing data are available in ENA repository under accession code PRJEB64781 and previously published RNA sequencing data used in Supplementary Fig. 2 are available in ENA repository under accession code PRJEB15150. Source data are provided with this paper.

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

## Acknowledgements

The authors would like to thank Ines Kleiber-Schaaf and Andrea Garcete-Bärtschi for their excellent support with histology; Caroline Mussak, Colin Sparano and Stanislav Dergun from the Institute of Experimental Immunology of the University of Zurich for their support with the FACS acquisition, Dr. Lennart Opitz from the Functional Genomics Center Zurich for support with the RNA sequencing. We would also like to thank Jan Käsler, Julia Maria Martinez Gomez and Mirka Schmid from the Division of Dermatology for the excellent support with the analysis of the samples from the treated melanoma patients. The imaging was performed with equipment maintained by the Center for Microscopy and Image Analysis, University of Zurich. Last but not least, we would like to thank all our patients that made this research possible. Funding: Sassella Foundation, grant numbers 19/15 to EG and 21/13 to SW, Research fellowship from the Academy of Finland to MH (no. 316340), ETH Zurich (Open ETH project SKINTEGRITY.CH to MD and CH), Novartis Foundation for Medical-Biological Research (Grant 22A038) and JOBST Award (German Society of Lymphology) to EG

## Author contributions

Conceptualization: E.G. Methodology: E.G., S.W., M.L., M.D., C.H. Investigation: S.W., M.M., P.T., M.H., S.T., J.v.A., R.D., N.L., E.G. Visualization: S.W., M.M., S.T., J.v.A., E.G. Funding acquisition: E.G., S.W., P.G., M.D., M.L., S.T. Project administration: E.G., S.W., R.D., M.L. Supervision: E.G. Writing – original draft: S.W., E.G. Writing – review & editing: S.W., M.M., P.T., M.H., S.T., J.v.A., P.G., R.D., N.L., C.H., M.D., M.L., E.G.

## Competing interests

NL acts as Scientific Advisor and Consultant for Medical Microinstruments (MMI). RD has intermittent, project-focused consulting and/or advisory relationships with Novartis, Merck Sharp & Dhome (MSD), Bristol-Myers Squibb (BMS), Roche, Amgen, Takeda, Pierre Fabre, Sun Pharma, Sanofi, Catalym, Second Genome, Regeneron, Alligator, T3 Pharma, MaxiVAX SA, Pfizer, Simcere and touchIME outside the submitted work. All other authors declare that they have no competing interests.
