## [Transparent Peer Review file · Nature Communications]

Anti-CTLA4 treatment reduces lymphedema risk potentially through a systemic expansion of the FOXP3+ Treg population

Corresponding Author: Dr Epameinondas Gousopoulos

Version 0:

Reviewer comments:

Reviewer #1

(Remarks to the Author)

This high-quality manuscript uses clinical data to indicate that anti-CTLA4 treatment reduces the risk of secondary lymphedema post-surgery for melanoma, and employs a mouse model to demonstrate that anti-CTLA4 restricts development of secondary lymphedema. The analysis of clinical data is a particular strength of this study, which is unusual in the lymphedema research field, and enhances the likelihood that clinical translation of an anti-CTLA4 approach could reduce the occurrence of lymphedema in patients after surgery for cancer. It would be interesting to know if an anti-CTLA4 approach could have any utility in treating pre-existing secondary lymphedema. Overall the manuscript is very well presented, the data are compelling, the claims are justified by the data and the findings constitute a significant step-forward for the field. Specific points which need to be addressed are listed below, most of which are minor in nature.

1. Lymphedema can take considerable, and highly variable, periods of time to develop post-surgery. It would therefore be important to know if the average time-period over which lymphedema had the opportunity to develop was similar for the five study groups in Fig. 1D&E. I could not find this information in the current version of the manuscript - it needs to be presented and/or discussed.
2. The Authors should comment on why there was no increase in FoxP3 mRNA even though there were many more cells expressing this protein (Fig 2B&D). Is there any evidence in the literature for translational regulation of FoxP3 synthesis?
3. In Results, the text refers to Figure 1D but I think this should be 2D (line 185).
4. There are mistakes in assigning statistical tests to panels in the legend to Fig. 3.
5. What statistical test was used in Fig. 5A?
6. Something has gone wrong on line 577 in Materials and Methods.

Reviewer #2

(Remarks to the Author)

Lymphedema is an insidiously troublesome condition for which we have no cure or truly effective treatment. A unpalatable feature of lymphedema is an inability to accurately predict who and when patients will develop this condition. The literature, from the bench side to the patient, has investigated the underpinnings of this disease, possible aspects of its etiology and how the disease itself can be prevented or mitigated either pharmacologically or surgically.

The paper by Gousopoulos and colleagues adds to the growing body of data on possible etiologic factors contributing to the development of lymphedema. The endeavor that was undertaken had its inception with a clinical observation that patients who received anti-CTLA4 treatment in situations of melanoma had a significant decrease in the clinical incidence of lymphedema. The clinical investigation went even further to truly define histopathologic differences to clarify the observed

clinical decrease in lymphedema. When matched for a number of clinical factors, an increased infiltration of CD4+, FOXP3+ and CTLA4+ cells were detected in lymphedema specimens and this histologic finding was confirmed with mRNA expression in the human specimens. The idea to investigate the genetic, molecular and cellular underpinnings of the clinical findings was done so using a widely accepted murine tail model of lymphedema. Here, the idea that lymphedema improvement is mediated through an anti-CTLA4-dependent Treg expansion was demonstrated through histologic evaluation, real time NIR lymphatic mapping, cellular phenotyping, cytokine, and genetic profiles.

I commend the authors on their insightful and thoughtful approach to this clinical observation and their diligence to tease apart the nuances of the clinical finding through use of a murine model of lymphedema. Again, this body of work greatly adds to the wealth of information that has already been investigated as it pertains to lymphedema. What adds weight to the validity of this study is that this investigation stemmed from a clinical observation on an already FDA approved drug that is widely used in a number of solid organ tumors.

The findings in this paper significantly add to what we already know about lymphedema from a molecular and cellular level. These findings add not only to our understanding but also help incite additional questions such as is this phenomenon observed in other solid organ tumors requiring lymphadenectomy receiving anti-CTLA4 treatment, or whether these findings are transient or long lasting.

The methods for data collection and analysis appear to have been completed at the highest level of fidelity with multiple modalities being employed to permit not only clarification of said results but to provide cross confirmation of the findings.

Presentation of the results is clear and concise and provides an easy avenue for the reader to follow.

Where I feel like clarification can be provided is in regard to the retrospective patient data. There remains a lack of a consensus on how to objectively and clinically determine lymphedema. In this paper there is a lack of transparency on what methods were used clinically to label patients as suffering from lymphedema. There are a number of clinical pathologic findings that indicate lymphedematous changes. Were limb measurements utilized and compared to the contralateral limb? If so were they done after a period of time following removal of compressive garments to allow for equalization of the limb volume? A more objective method that has been utilized in the office is NIR imaging. I think a clarification on the methods of lymphedema determination would add validity to this study. On a different note, were the various specialties, that being dermatology, plastic surgery or angiology, standardized in their determination of lymphedema? On a final note, I am curious how long after completion of the treatment did the patients on average develop signs and symptoms of lymphedema and what was the followup time for these patients? One feature of lymphedema is that the temporality of presentation can vary greatly to weeks and months to years after surgical insult.

Overall, this is an well thought out study that is comprehensive and exhaustive in its pursuit to clarify the clinical findings of anti-CTLA4 treatment.

Reviewer #3

(Remarks to the Author)

Thank you for the opportunity to review this paper. The authors aimed to assess anti-CTLA-4 treatment for the reduction of lymphoedema in a mouse model, to link with clinical observations in melanoma. They go on to attempt to link a presumed benefit with the expansion of regulatory T cells.

There may well be a benefit of clinical antiCTLA4 in reducing lymphoedema from the patient data, but the methodology is flawed and requires revisiting (see Major Point 1).

Experimentally, the effect of antiCTLA4 treatment is modest in mice and unclear whether it is sustained or simply due to a temporary modulation in inflammation. Specifically, in Figure 3 the authors measure tail vein volume up to week 2, showing a difference of around 200% tail volume to around 100% tail volume increase; yet in therapeutic studies tail volume is measured for much longer and there is a return to 0% in effective therapies. Critically, the current study does not show data beyond day 20 whereas most studies go to many weeks (e.g. Gardeneir et al Nat Comms 2017 goes to week 9). Moreover, most studies attempt to quantify lymphatic flow in the mouse tail model in more detail e.g. by assessing uptake of technitium99 in sacral lymph nodes; lymphatic vessel pulsation measurement; and provision of tail cross sectional histology (again see Gardeneir et al. Nat Comms 2017).

The link to Tregs is the weakest part of the study as it is largely observational with no information on mechanism provided, for example through experimental depletion of Tregs (e.g. using a Foxp3DTR mouse) to confirm the causality between antiCTLA-4 treatment, Treg expansion, and reduction of lymphoedema. Moreover it is not clear that there is an actual expansion in Tregs given the lack of cell count data. The transcriptomic data provide no further insight around mechanism.

1. Figure 1. It is crucial that the authors perform a regression analysis across all clinical and pathological (especially melanoma stage) characteristics to ensure that the result in Figure 1 is not secondary to another factor. For example, there appears to be an impact of sex on whether patients receiving LAD develop lymphoedema, with males perhaps more protected (Figure 1B). Does the limb affected, melanoma thickness, lymphatic involvement, metastatic spread etc impact the data? It is not appropriate to group all melanoma stages together. Moreover, please provide information on lymphoedema staging in these patients across the recognised 4 stage system in order to understand severity association with treatment,

given that lymphoedema is not simply binary.

2. Figure 2 is somewhat unhelpful as it compares pathology with no pathology, and of course pathology will have more cells in it. I am not sure that this is the correct control. A better comparison may be between severe and mild lymphoedematous tissue (e.g. stage 1 vs stage 4), for example between antiCTLA4 treated and untreated melanoma patients.

How were cells quantified in the biopsies? No information is presented in the methods and the concern is that it is manual and potentially biased. FOXP3 is not sufficient to identify Tregs in human biopsies due to its expression in effector cells. Multiplexed IF is needed for correct identification or a spatial profiling approach. The FOXP3 and CTLA4 staining images particularly are not convincing as the staining looks non-specific (e.g. CTLA4) there is therefore a need for technical control sections with no primary/no secondary antibodies, particularly as the CTLA4 staining is looking somewhat nuclear in the provided images. Why is there an increase in FOXP3+ cells on IHC yet no change by gene expression?

3. In Figure 3B it would be helpful to do an AUC comparison as well as repeated measures ANOVA. Importantly, tail vein volume monitoring must go on for much longer (a few weeks, see earlier comments), and more information on lymphatic function must be provided to ensure observed changes are actually secondary to enhanced lymphatic activity and not simply reduced inflammation. Similar to the point above, please provide methodological information quantification of cells by histology and ensure it is performed in an unbiased manner.

4. Figure 4 is overinterpreted. There is ostensibly no difference with treatment across anything measured, and the significance values are likely incorrect as these samples should not have been tested with separate t-tests given the multiple comparisons. These data should be presented with standard deviation.

5. Figure 5 is not meaningful without providing absolute cell counts as the percentage changes can represent an increase in only a few cells. Changes in percentage can be due to changes in other unmeasured cell populations. Tregs should be gated, at the very least, on CD25/Foxp3. Ideally this assay could have used a Foxp3-GFP mouse for correct Treg identification. The high CD127 expression across the presumed Treg population is concerning that these are not actually Tregs. Similarly, the statistical tests should probably not be separate t-tests given the multiple comparisons. The conclusion that treatment drives Foxp3 cell expansion is poorly substantiated by the data given the small sample size in Figure 5A, the large spread, and the lack of absolute cell counts. For the FACS plots, it would be preferable to see dot rather than contour plots to have better visualisation of cell counts.

6. In Figure 5 there are some anomalies in the data, in particular, why is there a large population of TCR+ double negative cells; why are there no CD8 cells; and why are 50% or more of the CD4+ cells expressing Foxp3 positive both in control and treated samples?

7. Figure 6G should show the data as paired and remove data points where there aren't post-treatment counts. As the data do not show absolute cell counts per volume blood, it is not possible to claim there has been an expansion in Tregs, as the observations may be due to a reduction in other CD4 cell types. Similar statistical problems here as previous figures, there should be statistical corrections for multiple comparisons.

Version 1:

Reviewer comments:

Reviewer #1

(Remarks to the Author)

The Authors have satisfactorily addressed all concerns I raised about the original version of the manuscript.

Reviewer #2

(Remarks to the Author)

I wanted to thank Dr. Gousopoulos and his team for kindly and concisely addressing my previous questions. I have found the answers to be more than acceptable.

I echo my previous assessment of their paper in stating that The paper by Gousopoulos and colleagues adds to the growing body of data on possible etiologic factors contributing to the development of lymphedema. The endeavor that was undertaken had its inception with a clinical observation that patients who received anti-CTLA4 treatment in situations of melanoma had a significant decrease in the clinical incidence of lymphedema. The clinical investigation went even further to truly define histopathologic differences to clarify the observed clinical decrease in lymphedema. When matched for a number of clinical factors, an increased infiltration of CD4+, FOXP3+ and CTLA4+ cells were detected in lymphedema specimens and this histologic finding was confirmed with mRNA expression in the human specimens. The idea to investigate the genetic, molecular and cellular underpinnings of the clinical findings was done so using a widely accepted murine tail model of lymphedema. Here, the idea that lymphedema

improvement is mediated through an anti-CTLA4-dependent Treg expansion was demonstrated through histologic evaluation, real time NIR lymphatic mapping, cellular phenotyping, cytokine, and genetic profiles.

I commend the authors on their insightful and thoughtful approach to this clinical observation and their diligence to tease apart the nuances of the clinical finding through use of a murine model of lymphedema. Again, this body of work greatly adds to the wealth of information that has already been investigated as it pertains to lymphedema. What adds weight to the validity of this study is that this investigation stemmed from a clinical observation on an already FDA approved drug that is widely used in a number of solid organ tumors.

The findings in this paper significantly add to what we already know about lymphedema from a molecular and cellular level. These findings add not only to our understanding but also help incite additional questions such as is this phenomenon observed in other solid organ tumors requiring lymphadenectomy receiving anti-CTLA4 treatment, or whether these findings are transient or long lasting. The methods for data collection and analysis appear to have been completed at the highest level of fidelity with multiple modalities being employed to permit not only clarification of said results but to provide cross confirmation of the findings.

Presentation of the results is clear and concise and provides an easy avenue for the reader to follow.

I have no further reservations about this publication.

Reviewer #3

(Remarks to the Author)

Thank you very much for your insightful responses and the additions made to the manuscript which are significant. Thank you also for thoroughly responding to my comments. I appreciate the efforts made to enhance the content, specifically the CD57BI/6 model and the inclusion of patient data. While the reference to previous studies that have explored Tregs in lymphoedema is appreciated, this does not necessarily prove mechanism definitively in the current study. I appreciate that it would be a challenge for the authors to perform further experiments to confirm this at this point, and would therefore rather the authors focus on my remaining points below.

1. Patient analysis: It is crucial that a proper multivariable logistic regression is performed to identify independent predictors of lymphoedema development. The analysis in Supplemental Table 3 does not provide this. A regression analysis will help ensure that the findings are robust and not confounded by other factors which on face value look to appear to be implicated. For example, patient sex is an issue, as well as, for example, site of melanoma - primarily occurring in the leg (63%) in patients with lymphoedema, while primarily occurring in the trunk in patients without lymphoedema.

2. Cell count data: The comment that PBMCs from a biobank preclude the provision of cell count data is noted with thanks. However, my question specifically relates to cell count data from the mouse studies, where these numbers are indeed available. This data could be presented graphically to clearly show increases in Treg numbers, not just ratios. The provision of the data only in the resource files for Figures 4 or 5 is insufficient for immediate/effective interpretation.

3. Figure 5A-B: The unusually low percentage of CD8 cells reported (between 2-10%) still needs further clarification. Why would this be an issue with flow compensation for only the CD8 T cells? Is the fluorochrome overlapping with Evans blue? If so perhaps this should be noted in the manuscript.

4. Figure 4: For clarity and to avoid confusion, I recommend that only the q-value, which accounts for the FDR, should be included in Figure 4. Presenting both p-value and q-value simultaneously is somewhat confusing to the reader.

5. Minor points:

- Supplementary Figure 3F: Please correct the labelling typo in "CTLA4".
- Figure 6: IL17A is not a canonical Treg cytokine. Could the authors please comment on this finding?
- Please change "alive cells" to "live cells".
- The title of Table 4 has a syntax error.

Version 2:

Reviewer comments:

Reviewer #3

(Remarks to the Author)

Many thanks for your thorough responses and revisions.

REVIEWER COMMENTS

Reviewer #1 (expert in lymphoedema):

This high-quality manuscript uses clinical data to indicate that anti-CTLA4 treatment reduces the risk of secondary lymphedema post-surgery for melanoma, and employs a mouse model to demonstrate that anti-CTLA4 restricts development of secondary lymphedema. The analysis of clinical data is a particular strength of this study, which is unusual in the lymphedema research field, and enhances the likelihood that clinical translation of an anti-CTLA4 approach could reduce the occurrence of lymphedema in patients after surgery for cancer. It would be interesting to know if an anti-CTLA4 approach could have any utility in treating pre-existing secondary lymphedema. Overall the manuscript is very well presented, the data are compelling, the claims are justified by the data and the findings constitute a significant step-forward for the field. Specific points which need to be addressed are listed below, most of which are minor in nature.

We want to thank Reviewer#1 for the very positive feedback and the helpful suggestions.

1. Lymphedema can take considerable, and highly variable, periods of time to develop post-surgery. It would therefore be important to know if the average time-period over which lymphedema had the opportunity to develop was similar for the five study groups in Fig. 1D&E. I could not find this information in the current version of the manuscript - it needs to be presented and/or discussed.

We appreciate the suggestion and agree that the period between surgery and diagnosis is an important parameter. Therefore, we reanalyzed our data and now provide this information in the manuscript (line 152-157) and Suppl. Table 4-6

Briefly, in our cohort a total of 68 patients developed lymphedema and the average time between lymphadenectomy and the diagnoses of lymphedema was 4.31 ± 7.05 months. The 40 patients who did not receive immunotherapy had an average time of 3.85 ± 7.60 months. The 28 patients receiving immunotherapy had an average time 4.96 ± 6.24 months and the analysis of the subgroups revealed an average time for anti-CTLA4 3 ± 0.00 months (2 patients), anti-PD1 4.91 ± 3.54 months (11 patients), the combination therapy of anti-CTLA4 and anti-PD1 group 7.33 ± 10.06 months (9 patients), and in the Others group 2.17 ± 1.47 months (6 patients). The comparison of all 5 groups revealed no significant difference among them.

2. The Authors should comment on why there was no increase in FoxP3 mRNA even though there were many more cells expressing this protein (Fig 2B&D). Is there any evidence in the literature for translational regulation of FoxP3 synthesis?

We thank the review for this suggestion. To our knowledge there is no evidence in the literature, but the Treg characterization was done in two different tissue types. Specifically, the histology analysis was done in skin tissue, whereas the qPCR was performed in fat tissue. This might explain the differences. We now communicate the tissue differences more clearly in the manuscript. (see line 190-191)

3. In Results, the text refers to Figure 1D but I think this should be 2D (line 185).

We apologize for that mistake, which we corrected accordingly.

4. There are mistakes in assigning statistical tests to panels in the legend to Fig. 3. We apologize for that mistake, which we corrected accordingly.

5. What statistical test was used in Fig. 5A?

To fulfill reviewers #3 requests, the experiment was repeated and the figure 5 was modified. 5A is now a FACS plot, but we have now included all statistical tests in the figure legend (line 387-389).

6. Something has gone wrong on line 577 in Materials and Methods.

We apologize for that mistake, which now has been corrected.

Reviewer #2 (expert in lymphoedema):

Lymphedema is an insidiously troublesome condition for which we have no cure or truly effective treatment. A unpalatable feature of lymphedema is an inability to accurately predict who and when patients will develop this condition. The literature, from the bench side to the patient, has investigated the underpinnings of this disease, possible aspects of its etiology and how the disease itself can be prevented or mitigated either pharmacologically or surgically.

The paper by Gousopoulos and colleagues adds to the growing body of data on possible etiologic factors contributing to the development of lymphedema. The endeavor that was undertaken had its inception with a clinical observation that patients who received anti-CTLA4 treatment in situations of melanoma had a significant decrease in the clinical incidence of lymphedema. The clinical investigation went even further to truly define histopathologic differences to clarify the observed clinical decrease in lymphedema. When matched for a number of clinical factors, an increased infiltration of CD4+, FOXP3+ and CTLA4+ cells were detected in lymphedema specimens and this histologic finding was confirmed with mRNA expression in the human specimens. The idea to investigate the genetic, molecular and cellular underpinnings of the clinical findings was done so using a widely accepted murine tail model of lymphedema. Here, the idea that lymphedema improvement is mediated through an anti-CTLA4-dependent Treg expansion was demonstrated through histologic evaluation, real time NIR lymphatic mapping, cellular phenotyping, cytokine, and genetic profiles.

I commend the authors on their insightful and thoughtful approach to this clinical observation and their diligence to tease apart the nuances of the clinical finding through use of a murine model of lymphedema. Again, this body of work greatly adds to the wealth of information that has already been investigated as it pertains to lymphedema. What adds weight to the validity of this study is that this investigation stemmed from a clinical observation on an already FDA approved drug that is widely used in a number of solid organ tumors.

The findings in this paper significantly add to what we already know about lymphedema from a molecular and cellular level. These findings add not only to our understanding but also help incite additional questions such as is this phenomenon observed in other solid organ tumors requiring lymphadenectomy receiving anti-CTLA4 treatment, or whether these findings are transient or long lasting.

The methods for data collection and analysis appear to have been completed at the highest level of fidelity with multiple modalities being employed to permit not only clarification of said results but to provide cross confirmation of the findings.

Presentation of the results is clear and concise and provides an easy avenue for the reader to follow.

We highly appreciate the comments of Reviewer #2 and want to thank for the opportunity to improve our manuscript.

Where I feel like clarification can be provided is in regard to the retrospective patient data. There remains a lack of a consensus on how to objectively and clinically determine lymphedema. In this paper there is a lack of transparency on what methods were used clinically to label patients as suffering from lymphedema. There are a number of clinical pathologic findings that indicate lymphedematous changes. Were limb measurements utilized and compared to the contralateral limb? If so were they done after a period of time following removal of compressive garments to allow for equalization of the limb volume? A more objective method that has been utilized in the office is NIR imaging. I think a clarification on the methods of lymphedema determination would add validity to this study. On a different note, were the various specialties, that being dermatology, plastic surgery or angiology, standardized in their determination of lymphedema?

We understand the reviewer's point of view and we are glad to increase the transparency of our manuscript by providing additional data in Table 4-6. In our study we explicitly included patients with a diagnosis confirmed both with an imaging methods such as MRI, CT, or sonography and by one of the specialists following up the patients, namely angiologists, dermatologists, or plastic surgeons. A detailed clinical examination was always undertaken, which did not always include documented measurements of the affected vs the contralateral limb. Therefore, such measurement can unfortunately not be provided. The fact of a difference in volume has always been documented in the clinical reports, along with other typical lymphedema characteristics, including the presence of pitting or non-pitting lymphedema and the dedicated imaging analysis. To further increase the transparency of the clinical characteristics of our patient cohort, we now added Suppl. Table 4, which displays by whom the diagnosis was performed and what objective imaging method was used for all lymphedema patients. In more detail, among the 68 patients with lymphedema after lymphadenectomy, the diagnosis was made by angiologists in 20% (14 patients) of cases, by dermatologists in 34% (23 patients), and by plastic surgeons in 46% (31 patients). In 43% (29 patients) of cases, the diagnosis was validated using MRI, in 29% (20 patients) using sonography, and in 28% (19 patients) using CT. Simultaneously, we also compared clinical manifestations. In 53% (36 of 68) of cases, there was pitting edema (Stage 2), and in 47% (32 of 68) of cases, there was non-pitting edema Stage 3. In patients with lymphedema in the lower limbs, 83% of cases (34 of 41) exhibited a positive Stemmer sign, while 17% of cases (7 of 41) showed a negative Stemmer sign.

On a final note, I am curious how long after completion of the treatment did the patients on average develop signs and symptoms of lymphedema and what was the follow-up time for these patients? One feature of lymphedema is that the temporality of presentation can vary greatly to weeks and months to years after surgical insult.

This is indeed a very valid point raised by the reviewer, as the onset of lymphedema appears variable among the patients receiving lymphadenectomy. All patients undergoing lymphadenectomy due to melanoma treatment, are regularly controlled by the dermatologists and the plastic surgeons. In particular, the dermatological assessment as part of the oncologic surveillance regime occurs every 3-6 months during the first 5 years. Patients with signs of lymphedema are referred to specialized plastic surgeons or angiologists for a further confirmation of the diagnosis.

The clinical criteria employed for the diagnosis of lymphedema are the same among all involved disciplines in our University Hospital. In order to faithfully answer the question raised by the reviewer, we re-examined the data to calculate the average time for the diagnosis of lymphedema after lymphadenectomy.

We now provide in the Suppl. Table 4 - 6 the time between lymphadenectomy (LAD) and lymphedema diagnosis of each individual patient. The average time required for the

diagnosis of lymphedema after LAD was 4.31 ± 7.05 months. The 40 patients who did not receive immunotherapy had an average time of 3.85 ± 7.60 months. The 28 patients receiving immunotherapy had an average time 4.96 ± 6.24 months and the analysis of the subgroups revealed an average time for anti-CTLA4 3 ± 0.00 months (2 patients), anti-PD1 4.91 ± 3.54 months (11 patients), the combination therapy of anti-CTLA4 and anti-PD1 group 7.33 ± 10.06 months (9 patients), and in the Others group 2.17 ± 1.47 months (6 patients).

These data are in line with current literature exhibiting lymphedema onset peaks 6-12 months upon the lymphatic vascular insult. (1)

Overall, this is a well thought out study that is comprehensive and exhaustive in its pursuit to clarify the clinical findings of anti-CTLA4 treatment.

We thank Reviewer #2 for such a very positive feedback.

Reviewer #3 (expert in Treg-based therapies):

Thank you for the opportunity to review this paper. The authors aimed to assess anti-CTLA-4 treatment for the reduction of lymphoedema in a mouse model, to link with clinical observations in melanoma. They go on to attempt to link a presumed benefit with the expansion of regulatory T cells.

There may well be a benefit of clinical antiCTLA4 in reducing lymphoedema from the patient data, but the methodology is flawed and requires revisiting (see Major Point 1). Experimentally, the effect of antiCTLA4 treatment is modest in mice and unclear whether it is sustained or simply due to a temporary modulation in inflammation. Specifically, in Figure 3 the authors measure tail vein volume up to week 2, showing a difference of around 200% tail volume to around 100% tail volume increase; yet in therapeutic studies tail volume is measured for much longer and there is a return to 0% in effective therapies. Critically, the current study does not show data beyond day 20 whereas most studies go to many weeks (e.g. Gardeneir et al Nat Comms 2017 goes to week 9). Moreover, most studies attempt to quantify lymphatic flow in the mouse tail model in more detail e.g. by assessing uptake of technitium99 in sacral lymph nodes; lymphatic vessel pulsation measurement; and provision of tail cross sectional histology (again see Gardeneir et al. Nat Comms 2017).

We thank Reviewer #3 for the critical review and the suggestions to improve our manuscript. A longer monitoring period would have been indeed desired, but as mentioned in lines 621-626 the use of the BALB/c mouse background is subjected to certain limitations. The BALB/c background mouse strain exhibits a prominent Th2-immune response and subsequently a stronger edema formation, which is beneficial to the methodology of our study and our effort to evaluate the potential of the anti-CTLA4 therapy in the lymphedema mouse-tail model. With increased lymphedema and naturally increased tail volumes, self-mutation occurred more frequently. This presents a clinical sign that poses a red flag in terms of animal welfare and is not tolerated by the respective veterinary authorities in Switzerland.

It is well documented in the literature that BALB/c mice have a significantly higher risk of complications after lymphedema inductions (2-5). For clarity reasons we provide here several publications, where these complications are clearly visible in the representative manuscript figures:

Example 1:

[Figure redacted]

Rutkowski et al. (2) Fig.3: Inflammation is visible at day 10.

Example 2:

[Figure redacted]

Uzarski et al (3) Figure 5: Bite marks and inflammation are visible on saline control day 10, VEGF-C day 10,15,20,25 and VEGF-C + VEGFR-3 day 5, 20 and 25

Example 3:

[Figure redacted]

Jin et al. (4) Figure 6: a clear wound on the distal side of the control tail is detectable.

Example 4:

[Figure redacted]

Jun et al. (5) Supplemental Fig. 5. Tail necrosis at 10 days postoperatively. Note the dry-up of the operated site (white arrows).

As described in line 627-629, our animal license that is reviewed by the cantonal veterinary office of Zurich includes clear exclusion criteria, such as “self-inflicted mutilation or severe abrasion the skin”. These complications were found to be significantly increased after 14 days, which made longer treatment period in this mouse strain incompatible with the Swiss animal welfare law.

In view of the skepticism of the reviewer and our efforts to provide more long-term treatment data, we repeated the experiment in C57BL/6J background mice and extended the treatment duration to 28 days. Using this mouse strain, we have been able to confirm once again the

protective effect of anti-CTLA4 against lymphedema. Given though the different immunological properties of this mouse-strain, the edema formation was, as expected, more moderate in comparison to the BALB/c background mice and respectively the protective effect of anti-CTLA4 was less prominent (but still significant). (see Suppl Figure 3 and lines 243-252 in the manuscript).

It is important to highlight that a 4-weeks monitoring period is in line with a large body of the existing literature, which investigated new lymphedema treatments. We summarized in Table 1 current pharmacological approaches used in the lymphedema mouse tail model.

Paper	Year	Treatment	Follow up	Volume changes (100% = volume prior surgery)
Nakamura et al. (6)	2009	Ketoprofen	11 days	Untreated 160% Ketoprofen 120%
Chang et al (7).	2013	Indomethacin, other enzyme inhibitors, antioxidants	14 days	Best treatment 190%
Zampell et al. (8)	2013	Anti IL4/ anti IL13 CD4 KO	42 days	Untreated 200% IL4 120% IL13 150% CD4 KO 125%
Jang et al. (9)	2016	Low-level laser therapy	14 days	
Gardenier et al.(10)	2017	Tacrolimus	63 days 21 days after therapy start	Untreated 150% Treatment start Week 3: 100% Week 6: 130%
Sakae et al(11)	2023	YIGSR peptide	35 days	Volume not provided, just increase in diameter.
Baiki et al(12)	2022	TGFb1 Antibody	28 days	Volume not provided, just increase in Skin thickness.
Wolf et al (current manuscript)	2024	aCTLA4	14 days 28 days	Control : 228% aCTLA4 :154% Control : 177% aCTLA4 :138%

Most studies had a follow up time ranging between 2 to 4 weeks. The study from Gardenier et al., which was explicitly mentioned by Reviewer 3, investigated the curative effect of tacrolimus. In fact, Gardenier et al focused on the therapy of established lymphedema and started the tacrolimus administration 3 or 6 weeks after surgery. **The follow-up time after the first administration of tacrolimus time was 3 weeks.** In contrast to the study pursued by Gardenier et al., the scope of the current approach is to **prevent** the onset of lymphedema and not to cure already established lymphedema. The additional study using the C57BL/6J mouse strain exhibits the favorable effect of anti-CTLA4 in the follow-up time of 4 weeks.

Furthermore, for the aforementioned studies including our current manuscript, we listed the final tail volume compared to the initial volume, in order to enable a direct comparison. A complete reduction to the initial tail volume of 100% was only detectable at the tacrolimus treatment started. To the best of our knowledge, we are not familiar with other

pharmaceutical studies against lymphedema where a return to the baseline (100% reduction) was achieved. All other studies mentioned above did not present an edema reduction back to the initial tail volume. In our view, the inability to achieve a complete return to the baseline is expected and reasonable, based on the animal model's pathophysiology and inherent limitations and is supported by the vast majority of the literature.

Regarding the assessment of lymphatic vascular function: we now provide a more in-depth analysis of the pumping activity of mouse tail (Figure 3F and suppl Figure 3F). These data indicate a significant improvement of the mean amplitude of the anti-CTLA4 treated mice, indicating an additional parameter to assess the improvement of the lymphatic vascular function. It is important to note that the radiotracer technetium-99m is not superior over the usage of the Near Infrared (NIR) dyes as outlined in several reviews (13). NIR dyes are a non-radioactive and cheaper alternative, which allows precise real time imaging. We believe that the additional assessment of the lymphatic function further strengthen our conclusions and we are thankful to the reviewer for requesting those.

The link to Tregs is the weakest part of the study as it is largely observational with no information on mechanism provided, for example through experimental depletion of Tregs (e.g. using a Foxp3DTR mouse) to confirm the causality between antiCTLA-4 treatment, Treg expansion, and reduction of lymphoedema. Moreover, it is not clear that there is an actual expansion in Tregs given the lack of cell count data. The transcriptomic data provide no further insight around mechanism.

We thank the reviewer for the additional suggestions. As a matter of fact, the senior author of this publication has previously published that the depletion of the Tregs using the Foxp3DTR mice results in aggravated lymphedema (14). Conversely, the expansion of the Tregs using the administration of IL-2/anti-IL-2 mAb complexes significantly reduced lymphedema development. Importantly, the therapeutic application of adoptively transferred Tregs upon lymphedema establishment reversed all of the major hallmarks of lymphedema, including edema, inflammation, and fibrosis, and also promoted lymphatic drainage function. These findings are published by Gousopoulos *et al.*(14) indicating the causality between Tregs and lymphedema development.

Regarding the transcriptomic data presented in our work, we do believe that these are indeed in line with recently published data, supporting our conclusions. In particular, the IFNbeta pathway, which is upregulated upon anti-CTLA4 treatment, was identified to restore both 15-LO expression and Treg cell number in a mouse model of lymphedema, as recently published in Nature Communications (15), thus matching our findings and conclusions. Regarding the absolute cell count data; as we received PBMCs from a biobank with variable viability, this also influences the absolute number of Tregs in the various samples. In order to follow the reviewer's suggestion and to exclude that the observed differences resulted from downregulation of other CD4 subgroups, we now always provide percentage of FOXP3+ CD25+ cells with all alive single cells. Furthermore, we show the absolute cell counts in the resource data file of Figure 4.

1. Figure 1. It is crucial that the authors perform a regression analysis across all clinical and pathological (especially melanoma stage) characteristics to ensure that the result in Figure 1 is not secondary to another factor. For example, there appears to be an impact of sex on whether patients receiving LAD develop lymphoedema, with males perhaps more protected (Figure 1B). Does the limb affected, melanoma thickness, lymphatic involvement, metastatic spread etc impact the data? It is not appropriate to group all melanoma stages together. Moreover, please provide information on lymphoedema staging in these patients across the recognised 4 stage system in order to understand severity association with treatment, given that lymphoedema is not simply binary.

We thank Reviewer #3 for the comment regarding the analysis of the retrospective patient cohort.

Further analysis on the variables mentioned by the reviewer has been now performed. As depicted in Suppl. Table 3, the Breslow and Clark levels as well as the other variables mentioned above have been analyzed in the lymphedema and non-lymphedema groups, without detecting any significant differences. We performed these analyses using Chi-Sq tests for categorical variables, and t-tests / ANOVA for continuous variables. We believe that these tests were the most appropriate for the data types, and hope that these tests satisfy the reviewer's suggestion.

Even though at first glance, it may appear that male sex may be linked to reduced lymphedema risk, our analysis indicates that the difference is not statistically significant (Suppl. Table 3).

We do agree with the reviewer that lymphedema stage is an interesting parameter and piece of information. We need to remain conscious though, that lymphedema is indeed not simply binary but a progressive condition that commonly aggravates over time in variable pace. Therefore, the staging of lymphedema is highly dependent on the monitoring time-period after the initial diagnosis. Following the suggestion of Reviewer #3 we added Suppl. Table 4 and provide now the stages of all 68 lymphedema patients and the categorization as pitting edema (Stage 2) and non-pitting edema (Stage 3). Additionally, and for reasons of clarity, we now provide the time between the lymphadenectomy (LAD) and the formal diagnosis of lymphedema.

2. Figure 2 is somewhat unhelpful as it compares pathology with no pathology, and of course pathology will have more cells in it. I am not sure that this is the correct control. A better comparison may be between severe and mild lymphoedematous tissue (e.g. stage 1 vs stage 4), for example between antiCTLA4 treated and untreated melanoma patients. How were cells quantified in the biopsies? No information is presented in the methods and the concern is that it is manual and potentially biased. FOXP3 is not sufficient to identify Tregs in human biopsies due to its expression in effector cells. Multiplexed IF is needed for correct identification or a spatial profiling approach. The FOXP3 and CTLA4 staining images particularly are not convincing as the staining looks non-specific (e.g. CTLA4) there is therefore a need for technical control sections with no primary/no secondary antibodies, particularly as the CTLA4 staining is looking somewhat nuclear in the provided images. Why is there an increase in FOXP3+ cells on IHC yet no change by gene expression?

We thank the reviewer for this suggestion. We do understand the view of the reviewer, but we do still believe that the comparison between lymphoedematous and healthy control tissue has its merits in this context. Of course, we agree that an analysis across all stages of lymphedema would be valuable, but it is unfortunately a rarity to have access to tissue from stage 1 lymphedema patients. The reason for this is that the tissue derives from lymphatic reconstructive surgical procedures and stage I patients do very rarely undergo such a procedure. To counterbalance the absence of stage I lymphedema tissue, we have now performed additional histological analysis of all patients included in our tissue-library and we present a stage-by-stage comparison for the available lymphedema stages. The majority of our patients exhibited Stage II lymphedema, while we were just able to collect two Stage 1 and one Stage 3 lymphedema patients. We now display these differences, without reaching any significance due to the low N number (see Figure 2).

Furthermore, we now provide a CD25 staining to ensure better evidence for the increased Tregs in the skin tissue. The CD25 staining replaces the staining of CTLA4 in Figure 2. In addition, we provide a representative FOXP3 (DAB=brown) and CD25 (magenta) double staining for additional clarity and the perusal of the reviewer:

FOXP3 (DAB=brown) and CD25 (magenta)

FOXP3 (DAB=brown)

CD25 (magenta)

negative control

Representative FOXP3 (DAB=brown) and CD25 (magenta) double staining of lymphedema tissue sections.

A CTLA4 staining is presented for demonstration along with the technical controls here:

Technical control without primary antibody with the CTLA4 staining protocol.

Technical control without primary antibody with the FOXP3 staining protocol.

Technical control without primary antibody with the CD25 staining protocol.

To ensure an unbiased quantification, the counting was done in an automated manner using QuPath and the procedure is now described in the materials and methods section.

The differences between FOXP3 expression and histology analysis could be explained with the fact that these analysis were performed in different tissues. While the histological analysis was performed in skin tissue, the qPCR was performed in fat tissue probes. We now communicate these tissue differences more clearly in the manuscript (see lines 190-191).

3. In Figure 3B it would be helpful to do an AUC comparison as well as repeated measures ANOVA. Importantly, tail vein volume monitoring must go on for much longer (a few weeks, see earlier comments), and more information on lymphatic function must be provided to ensure observed changes are actually secondary to enhanced lymphatic activity and not simply reduced inflammation. Similar to the point above, please provide methodological information quantification of cells by histology and ensure it is performed in an unbiased manner.

Following the reviewer's suggestions, we now provide an AUC comparison and a longer treatment duration with anti-CTLA4 (4 weeks) using the C57BL/6J mouse strain. These additional data are found in Suppl. Figure 3 and Suppl. Figure 4. Additionally, we now provide an enhanced analysis of the pumping activity in the mouse tail (Figure 3F and Suppl Figure 3E), indicating that not only increased amounts of dye are transferred into the imaged area but also the mean amplitude is increased upon anti-CTLA4 treatment, indicating an improved lymphatic transport capacity, thus further strengthening our hypothesis.

It is important to note that the scope of this work lies on the **onset** of lymphedema and the usage of anti-CTLA4 to prevent/diminish it. The long-term effect of the immune checkpoint inhibition using anti-CTLA4 becomes apparent in that the human part of the study and the animal models (despite their limitations versus the human condition and inherent restrictions they all naturally present) match the anticipated findings and confirm our reverse-translation findings.

Regarding the methodological approach used to quantify the number of cells in the histological slides, this was performed in this case manually but by two researchers blinded to the study, thus ensuring scientific integrity and an unbiased approach. We now provide this clarification in the materials and method section (lines xyz- xyz).

4. Figure 4 is overinterpreted. There is ostensibly no difference with treatment across anything measured, and the significance values are likely incorrect as these samples should not have been tested with separate t-tests given the multiple comparisons. These data should be presented with standard deviation.

We thank the reviewer for the helpful comment. To perform the analysis according to the expectations of the reviewer, we did reanalyze all our datasets. We now correct for multiple comparisons by controlling the False discovery rate using the method of Benjamini, Krieger and Yekutieli, providing the raw p-value and the q-value. Additionally, we now clearly mention in the results section, that only the p value is significant and that this only a trend supporting our hypothesis (see lines xyz). All values are presented with SD. These data are illustrated in the revised Figure 4.

5. Figure 5 is not meaningful without providing absolute cell counts as the percentage changes can represent an increase in only a few cells. Changes in percentage can be due to changes in other unmeasured cell populations. Tregs should be gated, at the very least, on CD25/Foxp3. Ideally this assay could have used a Foxp3-GFP mouse for correct Treg identification. The high CD127 expression across the presumed Treg population is concerning that these are not actually Tregs. Similarly, the statistical tests should probably not be separate t-tests given the multiple comparisons. The conclusion that treatment drives Foxp3 cell expansion is poorly substantiated by the data given the small sample size in Figure 5A, the large spread, and the lack of absolute cell counts. For the FACS plots, it would be preferable to see dot rather than contour plots to have better visualisation of cell counts.

We thank the reviewer for the comment, and we are glad to follow these suggestions. We repeated the FACS experiment and increased the size of the group to 7 mice per group. Treg are now identified using both FOXP3 and CD25 (CD25+FOXP3+). Furthermore, we now correct for multiple comparisons by controlling the False discovery rate using the method of Benjamini, Krieger and Yekutieli. Following the reviewer's suggestion we now provide dot plots for a better visualization, which are found in the revised Figure 5, further

confirming the increase of the CD25+FOXP3+ population within the CD4+ population. The absolute cell count numbers are provided in Source data file of figure5.

6. In Figure 5 there are some anomalies in the data, in particular, why is there a large population of TCR+ double negative cells; why are there no CD8 cells; and why are 50% or more of the CD4+ cells expressing Foxp3 positive both in control and treated samples?

We thank the reviewer for the comment. The discrepancies or anomalies that the reviewer notices are most probably an artifact, due to the inclusion of Evans blue, that is injected into the mouse tail to facilitate the lymphatic vascular ablation, resulting in complications with the compensation approaches.

To avoid such technical issues, we repeated the phenotypic characterization once again without the usage of Evans blue. The number of TCR+ double negative cells is reduced and CD8+ cells are now presented as well (see Fig. 5B).

We understand the concern of the reviewer regarding the percentage of Foxp3+ cells within the CD4+ compartment, which appeared quite high. It is important to consider though that the analysed skin, both in control and in treated mice is not normal skin but lymphedematous skin. While in "normal", non-lymphedematous skin tissue 7-8% Foxp3+ cells are present, in lymphedematous skin tissue the percentage of Foxp3 cells rises to 25-30% as already previously described (14). It is important to note, that the data in the aforementioned citation was generated in C57BL/6 mice, while the FACS analysis in our current manuscript is performed in BALB/c mice, which are known for their distinct immune cell composition and are reported to exhibit an even higher number of Foxp3 cells (16). Thus, the numbers of Foxp3+ cells presented are indeed within the expected range the lymphedema mouse tail model.

7. Figure 6G should show the data as paired and remove data points where there aren't post-treatment counts. As the data do not show absolute cell counts per volume blood, it is not possible to claim there has been an expansion in Tregs, as the observations may be due to a reduction in other CD4 cell types. Similar statistical problems here as previous figures, there should be statistical corrections for multiple corrections.

We are sorry to be unable to provide the absolute cell counts per volume of blood, as the isolated PBMCs derive from an established biobank and the requested information (regarding the original blood volume) is not available. Despite that, we managed to increase the number of patients included in our analysis and we now show only data in paired fashion. Additionally, the Tregs are identified as FOXP3+CD25+ cells.

What is more and in order to confirm the expansion of FOXP3+/CD25+ cells we now present the percentages of total alive single cells (Figure 6D) and corrected for multiple comparison analysis.

Despite not requested by the reviewer: In an effort to further strengthen the link between the anti-CTLA4 treatment and the Treg expansion, we additionally performed a paired analysis of the serum of the patients receiving anti-CTLA4, prior to the initiation of the treatment and at the end of the treatment period, focusing on Treg-related cytokines. The serum analysis of 16 patients could indeed strongly confirm our previous observations, as an increased expression of TNFa, IL10, IL17A and IL16 was found in the paired analysis. (Figure 6E and F).

Suppl. Table 4:

Pat. No	Treatment	Month from LAD to diagnosis	Diagnosis performed by	Lymphedema Stage	Diagnosis methode
1	No immunotherapy	1	Angiologist	Penoscrotal edema	Sonography
2	No immunotherapy	2	dermatologist	Stage 2	Sonography
3	No immunotherapy	2	Plastic surgeon	Stage 2	MRI
4	No immunotherapy	3	Dermatologist	Stage 2	Sonography
5	No immunotherapy	1	Dermatologist	Stage 2	Sonography
6	No immunotherapy	1	Plastic surgeon	Stage 2	CT
7	No immunotherapy	1	Plastic surgeon	Stage 3	MRI
8	No immunotherapy	4	Angiologist	Stage 3	CT
9	No immunotherapy	3	Plastic surgeon	Stage 3	MRI
10	No immunotherapy	8	Plastic surgeon	Stage 3	MRI
11	No immunotherapy	1	Plastic surgeon	Stage 3	CT
12	No immunotherapy	4	Dermatologist	Stage 2	CT
13	No immunotherapy	5	Dermatologist	Stage 3	MRI
14	No immunotherapy	1	Plastic surgeon	Stage 2	Sonography
15	No immunotherapy	2	Plastic surgeon	Stage 3	Sonography
16	No immunotherapy	3	Angiologist	Stage 3	CT
17	No immunotherapy	1	Dermatologist	Stage 3	MRI
18	No immunotherapy	5	Plastic surgeon	Stage 3	CT
19	No immunotherapy	1	Dermatologist	Stage 3	CT
20	No immunotherapy	1	Plastic surgeon	Stage 2	MRI

21	No immunotherapy	7	Dermatologist	Stage 2	CT
22	No immunotherapy	48	Angiologist	Stage 2	Sonography
23	No immunotherapy	2	Plastic surgeon	Stage 2	MRI
24	No immunotherapy	12	Angiologist	Stage 3	Sonography
25	No immunotherapy	1	Angiologist	Stage 2	Sonography
26	No immunotherapy	1	Plastic surgeon	Stage 2	MRI
27	No immunotherapy	2	Plastic surgeon	Stage 3	MRI
28	No immunotherapy	1	Plastic surgeon	Stage 3	MRI
29	No immunotherapy	1	Dermatologist	Stage 3	CT
30	No immunotherapy	1	Plastic surgeon	Stage 2	CT
31	No immunotherapy	1	Angiologist	Stage 3	Sonography
32	No immunotherapy	2	Plastic surgeon	Stage 2	MRI
33	No immunotherapy	1	Plastic surgeon	Stage 3	MRI
34	No immunotherapy	3	Plastic surgeon	Stage 2	CT
35	No immunotherapy	10	Angiologist	Stage 2	Sonography
36	No immunotherapy	3	Dermatologist	Stage 2	CT
37	No immunotherapy	1	Dermatologist	Stage 3	MRI
38	No immunotherapy	1	Angiologist	Stage 3	Sonography
39	No immunotherapy	3	Dermatologist	Stage 3	MRI
40	No immunotherapy	3	Angiologist	Stage 3	Sonography
41	Anti-CTLA4	3	Plastic surgeon	Stage 2	CT
42	Anti-CTLA4	3	Plastic surgeon	Stage 3	MR

43	Anti-PD1	2	Plastic surgeon	Stage 2	MRI
44	Anti-PD1	9	Angiologist	Stage 3	CT
45	Anti-PD1	8	Dermatologist	Stage 3	Sonography
46	Anti-PD1	6	Dermatologist	Stage 3	CT
47	Anti-PD1	7	Dermatologist	Stage 3	MRI
48	Anti-PD1	2	Angiologist	Stage 2	Sonography
49	Anti-PD1	11	Dermatologist	Stage 2	CT
50	Anti-PD1	1	Plastic surgeon	Stage 3	MRI
51	Anti-PD1	1	Plastic surgeon	Stage 2	MRI
52	Anti-PD1	5	Angiologist	Stage 2	Sonography
53	Anti-PD1	2	Plastic surgeon	Stage 2	MRI
54	Anti-PD1 + Anti-CTLA4	6	Plastic surgeon	Stage 3	MRI
55	Anti-PD1 + Anti-CTLA4	1	Dermatologist	Stage 3	Sonography
56	Anti-PD1 + Anti-CTLA4	1	Angiologist	Stage 3	MRI
57	Anti-PD1 + Anti-CTLA4	1	Plastic surgeon	Stage 3	MRI
58	Anti-PD1 + Anti-CTLA4	2	Plastic surgeon	Stage 3	CT
59	Anti-PD1 + Anti-CTLA4	15	Plastic surgeon	Stage 2	CT
60	Anti-PD1 + Anti-CTLA4	8	Dermatologist	Stage 2	CT
61	Anti-PD1 + Anti-CTLA4	31	Dermatologist	Stage 3	MRI
62	Anti-PD1 + Anti-CTLA4	1	Plastic surgeon	Stage 3	CT
63	others	1	Plastic surgeon	Stage 2	Sonography
64	others	4	dermatologist	edema	CT

65	others	1	Plastic surgeon	Stage 2	MRI
66	others	4	Dermatologist	Stage 3	MRI
67	others	1	Dermatologist	Stage 3	MRI
68	others	2	Dermatologist	Stage 2	MRI

Suppl. Table 6:

	NO immunotherapy	Anti-CTLA4	Anti-PD1	Anti-PD1 + Anti-CTLA4	Others
Patient No.	40	2	11	9	6
Time from LAD to diagnosis	3.85 ± 7.60	3 ± 0.00	4.91 ± 3.54	7.33 ± 10.06	2.17 ± 1.47
Edema Type					
Stage 2	18 (45,0%)	1 (50.0%)	6 (54.5%)	2 (22.2%)	4 (66.6%)
Stage 3	21 (52.5%)	1 (50.0%)	5 (45.5%)	7 (77.8%)	2 (33.3%)
Penoscrotal edema	1 (2.5%)				
Diagnosis by					
Plastic surgeon	18(45.0%)	2 (100%)	4 (36.4 %)	5 (55.6%)	2 (33.3%)
Angiologist	10 (25.0%)		3 (27.3%)	1 (11.1%)	
Dermatologist	12 (30.0%)		4 (36.4 %)	3 (33.3%)	4 (66.6%)
Diagnosis Methode					
CT	12 (30.0%)	1 (50.0%)	3 (27.3%)	4 (44.4%)	1 (16.7%)
Sonography	13 (32.5%)		3 (27.3%)	1 (11.1%)	1 (16.7%)
MRI	15 (37.5%)	1 (50.0%)	5 (45.5%)	4 (44.4%)	4 (66.6 %)

Suppl. Table 7:

	NO immunotherapy	immunotherapy
Patient No.	40	28
Time from LAD to diagnosis	3.85 ± 7.60	4.96 ± 6.24
Edema Type		
Stage 2	18 (45,0%)	13 (46.4 %)
Stage 3	21 (52.5%)	15 (53.6 %)
Penoscrotal edema	1 (2.5%)	
Diagnosis by		
Plastic surgeon	18(45.0%)	13 (46.4%)
Angiologist	10 (25.0%)	4 (14.3%)
Dermatologist	12 (30.0%)	11 (329.3%)
Diagnosis Methode		
CT	12 (30.0%)	9 (32.1%)
Sonography	13 (32.5%)	5 (17.9%)
MRI	15 (37.5%)	14 (50%)

1. McDuff SGR, Mina AI, Brunelle CL, Salama L, Warren LEG, Abouegylah M, et al. Timing of Lymphedema After Treatment for Breast Cancer: When Are Patients Most At Risk? *Int J Radiat Oncol Biol Phys.* 2019;103(1):62-70.
2. Rutkowski JM, Moya M, Johannes J, Goldman J, Swartz MA. Secondary lymphedema in the mouse tail: Lymphatic hyperplasia, VEGF-C upregulation, and the protective role of MMP-9. *Microvasc Res.* 2006;72(3):161-71.
3. Uzarski J, Drelles MB, Gibbs SE, Ongstad EL, Goral JC, McKeown KK, et al. The resolution of lymphedema by interstitial flow in the mouse tail skin. *Am J Physiol Heart Circ Physiol.* 2008;294(3):H1326-34.
4. Jin D, Harada K, Ohnishi S, Yamahara K, Kangawa K, Nagaya N. Adrenomedullin induces lymphangiogenesis and ameliorates secondary lymphoedema. *Cardiovasc Res.* 2008;80(3):339-45.
5. Jun H, Lee JY, Kim JH, Noh M, Kwon TW, Cho YP, et al. Modified Mouse Models of Chronic Secondary Lymphedema: Tail and Hind Limb Models. *Ann Vasc Surg.* 2017;43:288-95.
6. Nakamura K, Radhakrishnan K, Wong YM, Rockson SG. Anti-inflammatory pharmacotherapy with ketoprofen ameliorates experimental lymphatic vascular insufficiency in mice. *PLoS One.* 2009;4(12):e8380.
7. Chang TC, Uen YH, Chou CH, Sheu JR, Chou DS. The role of cyclooxygenase-derived oxidative stress in surgically induced lymphedema in a mouse tail model. *Pharm Biol.* 2013;51(5):573-80.
8. Zampell JC, Yan A, Elhadad S, Avraham T, Weitman E, Mehrara BJ. CD4(+) cells regulate fibrosis and lymphangiogenesis in response to lymphatic fluid stasis. *PLoS One.* 2012;7(11):e49940.
9. Jang DH, Song DH, Chang EJ, Jeon JY. Anti-inflammatory and lymphangiogenetic effects of low-level laser therapy on lymphedema in an experimental mouse tail model. *Lasers Med Sci.* 2016;31(2):289-96.
10. Gardenier JC, Kataru RP, Hespe GE, Savetsky IL, Torrisi JS, Nores GD, et al. Topical tacrolimus for the treatment of secondary lymphedema. *Nat Commun.* 2017;8:14345.
11. Sakae Y, Takada H, Ichinose S, Nakajima M, Sakai A, Ogawa R. Treatment with YIGSR peptide ameliorates mouse tail lymphedema by 67 kDa laminin receptor (67LR)-dependent cell-cell adhesion. *Biochem Biophys Res.* 2023;35:101514.
12. Baik JE, Park HJ, Kataru RP, Savetsky IL, Ly CL, Shin J, et al. TGF-beta1 mediates pathologic changes of secondary lymphedema by promoting fibrosis and inflammation. *Clin Transl Med.* 2022;12(6):e758.
13. Baeten IGT, Hoogendam JP, Braat A, Veldhuis WB, Jonges GN, Jurgenliemk-Schulz IM, et al. Fluorescent Indocyanine Green versus Technetium-99m and Blue Dye for Bilateral SENTinel Lymph Node Detection in Stage I-IIA Cervical Cancer (FluoreSENT): protocol for a non-inferiority study. *BMJ Open.* 2022;12(9):e061829.
14. Gousopoulos E, Proulx ST, Bachmann SB, Scholl J, Dionyssiou D, Demiri E, et al. Regulatory T cell transfer ameliorates lymphedema and promotes lymphatic vessel function. *JCI Insight.* 2016;1(16):e89081.
15. Zamora A, Nogue M, Verdu L, Balzan E, Draia-Nicolau T, Benuzzi E, et al. 15-Lipoxygenase promotes resolution of inflammation in lymphedema by controlling T(reg) cell function through IFN-beta. *Nat Commun.* 2024;15(1):221.

16. Godoy GJ, Paira DA, Olivera C, Breser ML, Sanchez LR, Motrich RD, et al. Differences in T regulatory cells between mouse strains frequently used in immunological research: Treg cell quantities and subpopulations in NOD, B6 and BALB/c mice. *Immunol Lett.* 2020;223:17-25.

REVIEWER COMMENTS

Reviewer #1 (Remarks to the Author):

The Authors have satisfactorily addressed all concerns I raised about the original version of the manuscript.

We would like to thank reviewer #1 for appreciating the outcome of the last revision and for approving the publication.

Reviewer #2 (Remarks to the Author):

I wanted to thank Dr. Gousopoulos and his team for kindly and concisely addressing my previous questions. I have found the answers to be more than acceptable.

I echo my previous assessment of their paper in stating that the paper by Gousopoulos and colleagues adds to the growing body of data on possible etiologic factors contributing to the development of lymphedema. The endeavor that was undertaken had its inception with a clinical observation that patients who received anti-CTLA4 treatment in situations of melanoma had a significant decrease in the clinical incidence of lymphedema. The clinical investigation went even further to truly define histopathologic differences to clarify the observed clinical decrease in lymphedema. When matched for a number of clinical factors, an increased infiltration of CD4+, FOXP3+ andCTLA4+ cells were detected in lymphedema specimens and this histologic finding was confirmed with mRNA expression in the human specimens. The idea to investigate the genetic, molecular and cellular underpinnings of the clinical findings was done so using a widely accepted murine tail model of lymphedema. Here, the idea that lymphedema improvement is mediated through an anti-CTLA4-dependent Treg expansion was demonstrated through histologic evaluation, real time NIR lymphatic mapping, cellular phenotyping, cytokine, and genetic profiles.

I commend the authors on their insightful and thoughtful approach to this clinical observation and their diligence to tease apart the nuances of the clinical finding through use of a murine model of lymphedema. Again, this body of work greatly adds to the wealth of information that has already been investigated as it pertains to lymphedema. What adds weight to the validity of this study is that this investigation stemmed from a clinical observation on an already FDA approved drug that is widely used in a number of solid organ tumors.

The findings in this paper significantly add to what we already know about lymphedema from a molecular and cellular level. These findings add not only to our understanding but also help incite additional questions such as is this phenomenon observed in other solid organ tumors requiring lymphadenectomy receiving anti-CTLA4 treatment, or whether these findings are transient or long lasting. The methods for data collection and analysis appear to have been completed at the highest level of fidelity with multiple modalities being employed to permit not only clarification of said results but to provide cross confirmation of the findings.

Presentation of the results is clear and concise and provides an easy avenue for the reader to follow.

I have no further reservations about this publication.

We would like to thank Reviewer #2 for the utterly positive feedback and the approval of our manuscript following the last revision.

Reviewer #3 (Remarks to the Author):

Thank you very much for your insightful responses and the additions made to the manuscript which are significant. Thank you also for thoroughly responding to my comments. I appreciate the efforts made to enhance the content, specifically the CD57Bl/6 model and the inclusion of patient data. While the reference to previous studies that have explored Tregs in lymphoedema is appreciated, this does not necessarily prove mechanism definitively in the current study. I appreciate that it would be a challenge for the authors to perform further experiments to confirm this at this point, and would therefore rather the authors focus on my remaining points below.

We thank Reviewer #3 for the appreciation of our efforts to thoroughly address all concerns raised during the previous revision period.

1. Patient analysis: It is crucial that a proper multivariable logistic regression is performed to identify independent predictors of lymphoedema development. The analysis in Supplemental Table 3 does not provide this. A regression analysis will help ensure that the findings are robust and not confounded by other factors which on face value look to appear to be implicated. For example, patient sex is an issue, as well as, for example, site of melanoma - primarily occurring in the leg (63%) in patients with lymphodema, while primarily occurring in the trunk in patients without lymphodema.

We thank the reviewer for the suggestion. We fit a multivariate generalized linear model (logistic link) to assess the influence of age, sex, Clark-level, Breslow thickness, localisation, presence of ulceration, primary diagnosis, and immunotherapy (yes/no) and the outcome is presented in Supplemental Table 3. Of all these variables, only immunotherapy had a significant influence on the probability of developing lymphedema. We note that the R function “anova” calculates p-values using type I sums of squares, in which the deviance associated with each variable is modeled sequentially according to their position in the model specification. Therefore, as we specify immunotherapy to be the last in the list of variables examined, we see that it is significantly associated with lymphedema even when possible confounding effects of the other variables removed. Further, none of them were significantly associated with lymphedema themselves when considered in a multivariate model. See below and Supplementary Table 3:

	NO Lymphedema	Lymphedema	p-value
n	411	68	
Age (mean (SD))	63.30 ± 14.58	62.63 ± 14.20	0.446
Sex = M (%)	252 (61.3 %)	33 (48.5 %)	0.727
Breslow (mean (SD))	3.61 ± 3.26	2.96 ± 2.78	0.265
Clark.Level (mean (SD))	3.85 ± 0.69	3.84 ± 0.69	0.356
Diagnosis (%)			0.597
Cerebral metastatic melanoma	41 (10.0 %)	3 (4.4 %)	
Lymphoid metastatic melanoma	0 (0.0 %)	1 (1.5 %)	
Metastatic melanoma	315 (76.6 %)	52 (76.5 %)	

Primary cutaneous melanoma	55 (13.4 %)	12 (17.6 %)	
Localization (%)			0.486
	21 (5.1 %)	4 (5.9 %)	
arm	77 (18.7 %)	10 (14.7 %)	
foot left	1 (0.2 %)	0 (0.0 %)	
head	38 (9.2 %)	2 (2.9 %)	
leg	123 (29.9 %)	43 (63.2 %)	
trunk	151 (36.7 %)	9 (13.2 %)	
Ulceration (%)	104 (28.8 %)	12 (20.3 %)	0.738
Immunotherapy	238 (89.5%)	28 (10.5%)	0.0005

2. Cell count data: The comment that PBMCs from a biobank preclude the provision of cell count data is noted with thanks. However, my question specifically relates to cell count data from the mouse studies, where these numbers are indeed available. This data could be presented graphically to clearly show increases in Treg numbers, not just ratios. The provision of the data only in the resource files for Figures 4 or 5 is insufficient for immediate/effective interpretation.

We thank the reviewer for the clarification. We now provide a graphical presentation of the absolute T_{reg} cell counts in Supplementary Figure 5.

3. Figure 5A-B: The unusually low percentage of CD8 cells reported (between 2-10%) still needs further clarification. Why would this be an issue with flow compensation for only the CD8 T cells? Is the fluorochrome overlapping with Evans blue? If so perhaps this should be noted in the manuscript.

The point raised by the reviewer is legitimate and the percentage of CD8 cells appears lower. In the first version of the manuscript, we had in indeed an issue fluorescence of Evans Blue, but in the current FACS analysis was performed without Evans Blue to minimize any overlap with the used fluorochromes. As mentioned before, the immunological niche in edematous tissue is very unique: A recent publication from Zamora et al in Nature Communications showed 1,2% - 2,0% CD8+ cells within the CD45+ population in edematous mouse tissue ¹. Gousopoulos et al. detected “just” 0.04% CD8+ cells of all single cells compared to 0.5% CD4+ cells ². The 0.5%-3% CD8+ cells (within the CD45+ cell population) in this study is in line with already published results.

4. Figure 4: For clarity and to avoid confusion, I recommend that only the q-value, which accounts for the FDR, should be included in Figure 4. Presenting both p-value and q-value simultaneously is somewhat confusing to the reader.

We thank the reviewer for this clarification. While we do see the merits of presenting both the p-value and the q-value, we have decided to follow the suggestion of the reviewer and remove the p-value from Figure 4.

5. Minor points:

- Supplementary Figure 3F: Please correct the labelling typo in "CTLA4".

We corrected typo accordingly.

- Figure 6: IL17A is not a canonical Treg cytokine. Could the authors please comment on this finding?

We thank the reviewer for the remark. Indeed, the pro-inflammatory cytokine IL17A is not a canonical Treg cytokine. Over the last year though, there is growing evidence that Tregs are able to produce IL17A ³. In our manuscript, the increased levels of IL17A cannot be solely attributed to the Treg expansion, but CTLA4 blockage has been found to be associated with an induction of IL17A-secreting CD4 T cells ⁴.

- Please change “alive cells” to “live cells”.

We corrected all figures accordingly.

- The title of Table 4 has a syntax error.

Thank you for the precise revision.

- 1 Zamora, A. *et al.* 15-Lipoxygenase promotes resolution of inflammation in lymphedema by controlling T(reg) cell function through IFN-beta. *Nat Commun* **15**, 221, doi:10.1038/s41467-023-43554-y (2024).
- 2 Gousopoulos, E., Proulx, S. T., Scholl, J., Uecker, M. & Detmar, M. Prominent lymphatic vessel hyperplasia with progressive dysfunction and distinct immune cell infiltration in lymphedema. *Am J Pathol* **186**, 2193-2203, doi:10.1016/j.ajpath.2016.04.006 (2016).
- 3 Jung, M. K., Kwak, J. E. & Shin, E. C. IL-17A-Producing Foxp3(+) Regulatory T Cells and Human Diseases. *Immune Netw* **17**, 276-286, doi:10.4110/in.2017.17.5.276 (2017).
- 4 von Euw, E. *et al.* CTLA4 blockade increases Th17 cells in patients with metastatic melanoma. *J Transl Med* **7**, 35, doi:10.1186/1479-5876-7-35 (2009).